# Continuous Field Reconstruction from Sparse Observations with Implicit Neural Networks

**Xihaier Luo, Wei Xu, Yihui Ren, Shinjae Yoo**
Brookhaven National Laboratory
{xluo,xuw,yren,sjyoo}@bnl.gov

**Balasubramanya Nadiga**
Los Alamos National Laboratory
{balu}@lanl.gov

## Abstract

Reliably reconstructing physical fields from sparse sensor data is a challenge that frequently arises in many scientific domains. In practice, the process generating the data often is not understood to sufficient accuracy. Therefore, there is a growing interest in using the deep neural network route to address the problem. This work presents a novel approach that learns a continuous representation of the physical field using implicit neural representations (INRs). Specifically, after factorizing spatiotemporal variability into spatial and temporal components using the separation of variables technique, the method learns relevant basis functions from sparsely sampled irregular data points to develop a continuous representation of the data. In experimental evaluations, the proposed model outperforms recent INR methods, offering superior reconstruction quality on simulation data from a state-of-the-art climate model and a second dataset that comprises ultra-high resolution satellite-based sea surface temperature fields.[Project Website: data & code]

## 1 Introduction

Achieving accurate and comprehensive representation of complex physical fields is pivotal for tasks spanning system monitoring and control, analysis, and design. However, in a multitude of applications, encompassing geophysics (Reichstein et al., 2019), astronomy (Gabbard et al., 2022), biochemistry (Zhong et al., 2021), fluid mechanics (Deng et al., 2023), and others, using a sparse sensor network proves to be the most practical and effective solution. In meteorology and oceanography, variables such as atmospheric pressure, temperature, salinity/humidity, and wind/current velocity must be reconstructed from sparsely sampled observations.

Currently, two distinct approaches are used to reconstruct full fields from sparse observations. Traditional physics model-based approaches are based on partial differential equations (PDEs). These approaches draw upon theoretical techniques to derive PDEs rooted in conservation laws and fundamental physical principles (Hughes, 2012). Yet, in complex systems such as weather (Brunton et al., 2016) and epidemiology (Massucci et al., 2016), deriving comprehensive models that are both sufficiently accurate and computationally efficient remains elusive. Moreover, integrating field data into these derived PDEs for validation and calibration poses significant challenges (Raissi et al., 2019). Concurrently, machine-learning-based approaches emerge as an alternative avenue for nonlinear field reconstruction (Mescheder et al., 2019; Sitzmann et al., 2020; Mildenhall et al., 2021).

In contrast to standard image and video data, scientific data describing complex physical systems present unique challenges. For example, sparse seismic networks (*sparse spatial coverage*) can lead to smaller earthquakes being unnoticed or their epicenters being misestimated (Myers & Schultz, 2000). Meanwhile, fluid dynamics in turbulent flows, an example of *high nonlinearity*, exhibit nonlinear behavior due to interactions between vortices and eddies (Stachenfeld et al., 2021). Other examples include *sensor mobility*, e.g., ocean waves and currents that transport floating buoys (Rodrigues et al., 2021), and *on-off dynamics*, e.g., cloud cover impacting solar panels that cause power fluctuations and grid instability (Paletta et al., 2022). These factors are driving the advancement of novel machine learning models, aiming to enhance and refine current approaches for field reconstruction. In this work, we introduce the first implicit neural representation (INR)-based model for global field reconstruction of scientific data from sparse observations with the following contributions:

• We introduce a context-aware indexing mechanism that compared to standard time index ($t$)-based INR models, incorporates additional semantic information.

• The presented network factorizes target signals into a set of multiplicative basis functions, subsequently applying element-wise shift and scale transformations to amalgamate latent information.

• Empirical validation demonstrates the proposed model achieves an average relative error reduction of $39.19\%$ compared to other state-of-the-art INR models.

## 2 RELATED WORK

### 2.1 CLASSICAL METHODS

**Regression Methods.** Field reconstruction resembles regression, predicting new outcomes at new locations. The most straightforward approach is to construct a linear model using available data. To account for nonlinearity, inverse distance weighting (IDW) employs distance weighting with a power parameter for weighted averaging in spatial interpolation (Shepard, 1968). While popular, IDW assumes isotropy and operates within the convex hull of data points. A more potent alternative is Gaussian Process (Rasmussen et al., 2006). In practice, the application of Gaussian Process can be hindered by the computational complexity of inverting the covariance matrix, which scales with a time complexity of $\mathcal{O}(n^3)$ and renders it infeasible for extremely large datasets (Angell & Sheldon, 2018; Yadav et al., 2021).

**Model Reduction Methods.** Model reduction techniques address the reconstruction task by converting the continuous spatial representation $u$ into a composite of basis functions, often referred to as *modes*: $u(\boldsymbol{x}) \approx \hat{u}(\boldsymbol{x}) = \sum_{i=1}^{m} a_i \phi_i(\boldsymbol{x})$. Coefficients $a_i$ and modes $\phi_i(\boldsymbol{x})$ typically are determined through optimization or regression models. Extensions of existing model reduction techniques for reconstructing full-field from partial-field measurements typically incorporate a mask function $m(u, \boldsymbol{x})$ that is defined as 0 where data are missing and 1 where data are present, such as in the case of Gappy proper orthogonal decomposition (Everson & Sirovich, 1995; Bui-Thanh et al., 2004). Recently, deep learning techniques have been employed to construct a nonlinear manifold, enhancing model performance for slowly decaying Kolmogorov $n$-width problems (Lee & Carlberg, 2020; Lusch et al., 2018; Kim et al., 2019).

### 2.2 DEEP LEARNING METHODS

**Super Resolution (SR).** SR typically focuses on upscaling low-resolution images $u_{low}$ to higher resolution $u_{high}$. Yet in the context of field reconstruction, we lack such paired data. Our primary focus is on generating a continuous representation $u$ from a discretized dataset. Recent progress in deep-learning-based SR enables continuous magnification through diverse techniques, including *innovative training* approaches, such as scale-consistent positional encodings (Ntavelis et al., 2022) and variable-size training (Chai et al., 2022); *local conditioning* methods that use deep surrounding features (Chen et al., 2021) or neighborhood-based interpolation (Luo et al., 2023); and *global conditioning* methods that leverage continuous coordinates and latent variables, e.g., MeshfreeFlowNet (Esmaeilzadeh et al., 2020) and Neural Implicit Flow (NIF) (Pan et al., 2023).

**Neural Inpainting.** Image inpainting techniques can be broadly classified into two categories: *traditional* and *learning-based* methods. *Traditional* methods primarily rely on low-level features, employing approaches like diffusion- (Bertalmio et al., 2000) or patch-based (Barnes et al., 2009) methods to extend information from surrounding regions into the missing areas. On the other hand, *learning-based* methods, particularly those employing GANs (Yu et al., 2018; Lee et al., 2020) and probabilistic diffusion models (Song et al., 2023b; Chung et al., 2023), have achieved more precise and semantically meaningful inpainting results. However, these *learning-based* methods may require post-processing and can be computationally demanding.

**Implicit Neural Representations.** INR models rely on coordinate-based neural networks (Xie et al., 2022). INRs can serve two main purposes: they either parameterize the *sensor* domain or the *density* domain directly. In the former case, INRs map sensor coordinates to predicted sensor activations, which can be used to enhance real measurement data. For the latter, INRs directly predict the density value at a two- or three-dimensional (2D/3D) spatial coordinate. Typically, raw sensor mea-

surements are derived from spatially varying density using transformations. Therefore, such direct prediction is supervised by mapping the model's output back to the sensor domain through various transformations, such as Radon in computed tomography (Reed et al., 2021), Fourier in magnetic resonance imaging (Song et al., 2023a), or convolution in cryo-electron microscopy (Zhong et al., 2021).

## 3  METHODOLOGY

**Overview.** The objective is to accurately reconstruct a spatiotemporal continuous physical field, denoted as $\boldsymbol{u}$, representing quantities such as temperature, velocity, or displacement. This field $\boldsymbol{u}$ is inherently a function of both spatial coordinates ($\boldsymbol{x}$) and time ($t$). Directly modeling such complex spatiotemporal physical fields poses significant challenges. Consequently, methodologies, e.g., functional separation of variables (Donà et al., 2021), have been devised to mitigate complexity, enhance tractability, and improve physical interpretability. Using these approaches, the underlying physical process $\boldsymbol{u}(\boldsymbol{x}, t)$ can be decomposed, for example, in the form of the product as $f_1(\boldsymbol{x}) \cdot f_2(t)$.

**Proposed Method.** Conventional spatiotemporal disentangled representation utilizes the time index ($t$) primarily as a reference to indicate a specific time instance. Motivated by the desire for a more context-aware indexing mechanism, we pose the question: *Can the pointing process be improved?* A natural approach to incorporate available context information in field reconstruction involves using measurements of the underlying physical process at time $t$. As the number and positions of available measurements change over time, we propose a design wherein an encoder extracts a latent representation from actual measurements. This latent representation is subsequently employed to guide the model to the target time instance. When coupled with an INR-based decoder, this proposed method achieves continuous field reconstruction. Figure 1 provides a comparison between the conventional scalar-index-based INR and our context-aware INR.

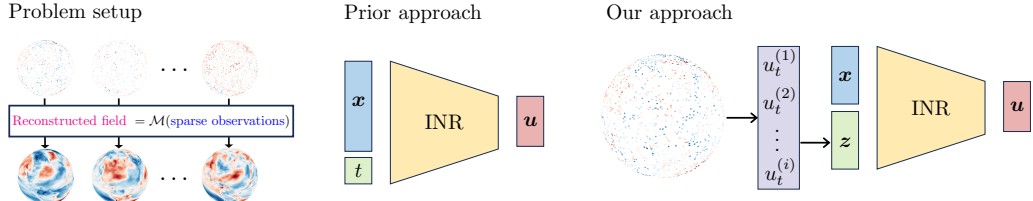

Figure 1: Field reconstruction from sparse observations: The *Prior approach* uses the time index ($t$) as a reference to indicate a specific time instance. *Our approach* is context-aware, leveraging available context information by incorporating measurements at time $t$.

**Overall Architecture.** We introduce a neural network reconstruction method, *MMGN* (Multiplicative and Modulated Gabor Network), that features an encoder-decoder architecture. The encoder extracts features from available measurements $\mathcal{U}_t = \{u_t^{(1)}, u_t^{(2)}, \dots\}$ at time $t$, while the decoder, guided by spatial coordinates $\boldsymbol{x}$ and a context-aware latent code $\boldsymbol{z}_t$, performs inference for the specific point at time $t$ and location $\boldsymbol{x}$. Overall, the model is defined in Equation (1).

$$\boldsymbol{z}_t = E_\varphi(\mathcal{U}_t), \quad \hat{u}(\boldsymbol{x}, t) = D_\phi(\boldsymbol{z}_t, \boldsymbol{x}), \quad \forall t \in \mathcal{T} \ \text{ and } \ \forall \boldsymbol{x} \in \Omega, \tag{1}$$

where $\Omega \subset \mathbb{R}^{d_{\boldsymbol{x}}}$ and $\mathcal{T} = \{t_i \in \mathbb{R}_+\}_{i=1}^{N_t}$ are the spatial and temporal domains, respectively.

### 3.1  ENCODER

Autoencoders (AE) and their probabilistic version, variational autoencoders (VAE) (Kingma & Welling, 2014), are commonly used for representation learning due to their natural latent variable formations. However, vanilla AE and VAE struggle with the randomness of sensor locations and numbers over time. While their graph counterparts handle spatial randomness and modified versions manage dynamic graphs, they become computation-intensive for large graphs and struggle with long-range dependencies (Pfaff et al., 2021). In contrast to AE and VAE, auto-decoder, as

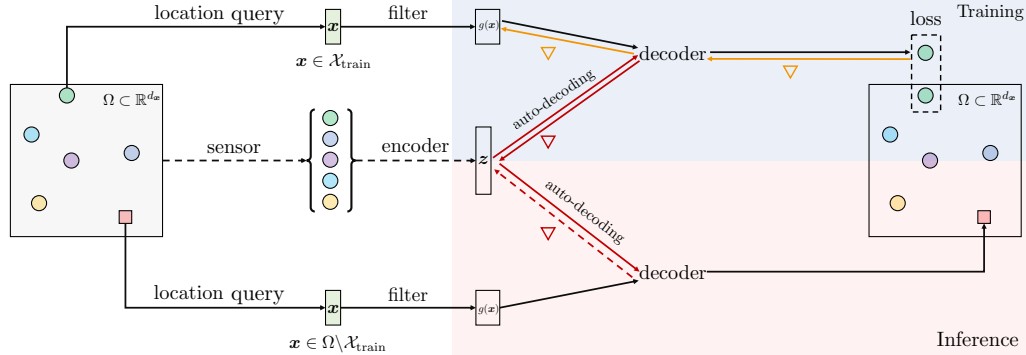

Figure 2: Architecture of the MMGN Model. The MMGN model employs auto-decoding to infer the latent variable $\boldsymbol{z}$. Consequently, only the decoder is explicitly defined, and encoding takes place through stochastic optimization. More precisely, the latent code $\boldsymbol{z} = \arg\min_{\boldsymbol{z}} \mathcal{L}(\boldsymbol{z}, \Theta)$ is obtained by minimizing a loss function $\mathcal{L}$ calculated as an expectation over a dataset.

demonstrated in Park et al. (2019), exhibits reduced underfitting and enhanced flexibility. It accommodates free-formed observation grids, including irregular ones or those on a manifold, without necessitating a specialized encoder architecture—as long as the decoder possesses the same property.

**Auto-decoder.** The aim of an *auto-decoder* is to compress essential information into $\boldsymbol{z}_t$, enabling the reconstructed value $\hat{u}_t^{(i)}$ to closely approximate the original value $u_t^{(i)}$ for any point within the domain. This is accomplished through an iterative process $\boldsymbol{z}_t^{(0)} \to \boldsymbol{z}_t^{(1)} \to \ldots$, employing gradient descent optimization. To initialize the trainable latent codes, we can assume the prior distribution over codes $p(\boldsymbol{z})$ follow a zero-mean multivariate Gaussian with a spherical covariance $\sigma^2 I$ (Xie et al., 2022). In practice, we empirically notice that initializing $\boldsymbol{z}_t^{(0)}$ to 0 yields slightly better results than Gaussian initialization:

$$\boldsymbol{z}_t^{(0)} = 0; \quad \boldsymbol{z}_t^{(i+1)} = \boldsymbol{z}_t^{(i)} - \alpha \bigtriangledown_{\boldsymbol{z}_t} \mathcal{L}(\hat{u}_t^{(i)}, u_t^{(i)}) \quad \text{for} \quad i = 0, \ldots, N-1, \tag{2}$$

where $\alpha$ is the learning rate, $N$ is the number of iteration steps, and $\mathcal{L}(\cdot)$ is the loss function.

## 3.2 DECODER

The decoder inputs consist of two parts: spatial coordinates $\boldsymbol{x}$ and latent codes $\boldsymbol{z}$. Subjecting $\boldsymbol{x}$ to fully connected feed-forward layers yields a coordinate-based multilayer perceptron (MLP). While such a coordinate-based MLP can offer a continuous representation, it struggles to learn high-frequency signals, a phenomenon known as *spectral bias*. Recent research indicates this issue can be mitigated using positional encoding with Fourier features (Tancik et al., 2020) or periodic nonlinearities in the first hidden layer (Sitzmann et al., 2020). In lieu of Fourier bases, we use Gabor filters to transform the coordinates as Fourier transforms emphasize a global frequency representation, making them less suitable for capturing varying frequency and orientation across different parts of the signal and more susceptible to noise and amplitude fluctuations. Specifically, we employ $N_g$ shift-invariant Gabor filters in the following form:

$$g_i(\boldsymbol{x}) = \exp\left(-\frac{\boldsymbol{\gamma}^{(i)}}{2} \left\| \boldsymbol{x} - \boldsymbol{\mu}^{(i)} \right\|_2^2\right) \sin\left(\boldsymbol{W}_g^{(i)} \boldsymbol{x} + \boldsymbol{b}_g^{(i)}\right), i = 1, \ldots, N_g, \tag{3}$$

where $\boldsymbol{\mu}^{(i)} \in \mathbb{R}^{d_h}$ and $\boldsymbol{\gamma}^{(i)} \in \mathbb{R}^{d_h \times d_x}$ denote the respective mean and scale term of the $i$th Gabor filter. The former is associated with the central frequency of the sinusoidal waveform that $g_i(\boldsymbol{x})$ is designed to identify, while the latter parameter corresponds to the standard deviation of the Gaussian envelope that modulates the sinusoidal waveform. This filter exhibits a multiplicative property (Fathony et al., 2020), which implies the product of the outcomes can be written from any pair

of Gabor filters $g_1(\boldsymbol{x}), g_2(\boldsymbol{x})$ into a summation of Gabor bases $g_1(\boldsymbol{x}) \circ g_2(\boldsymbol{x}) = \sum_{i=1}^{N} \beta_i g_i(\boldsymbol{x})$ with $\beta_{1:N}$ denoting the coefficients. This decomposability property facilitates the construction of hierarchical features, allowing the model to capture different levels of abstraction in the data and enhancing the interpretability of learned features.

After transforming the coordinates $g(\boldsymbol{x})$, we introduce a modulation step where the transformed coordinates are modulated through a multiplicative layer, thereby integrating $\boldsymbol{x}$ and $\boldsymbol{z}$.

**Multiplicative Filter Network.** Similar to the multiplicative filter network approach (Fathony et al., 2020), the modulation layer involves the iterative application of nonlinear Gabor filters to the network's input. These filters are then multiplied by the linear transformations of both $\boldsymbol{x}$ and $\boldsymbol{z}$. Specifically, in a decoder comprising $L$ layers, the decoding process is defined iteratively as follows:

$$
\begin{aligned}
\boldsymbol{h}^{(1)} &= g_1(\boldsymbol{x}) \\
\boldsymbol{h}^{(i+1)} &= g_i(\boldsymbol{x}) \odot \left( \boldsymbol{W}_{\boldsymbol{h}}^{(i)} \boldsymbol{h}^{(i)} + \boldsymbol{W}_{\boldsymbol{z}}^{(i)} \boldsymbol{z} + \boldsymbol{b}_{\boldsymbol{h}}^{(i)} \right), i = 1, \ldots, L-1 \\
D_\phi(\boldsymbol{z}, \boldsymbol{x}) &= \boldsymbol{W}_{\boldsymbol{h}}^{(L)} \boldsymbol{h}^{(L)} + \boldsymbol{b}_{\boldsymbol{h}}^{(L)},
\end{aligned}
\tag{4}
$$

where $\boldsymbol{W}_{\boldsymbol{h}}^{(i)} \in \mathbb{R}^{d_{i+1} \times d_i}$, $\boldsymbol{W}_{\boldsymbol{z}}^{(i)} \in \mathbb{R}^{d_{i+1} \times d_{\boldsymbol{z}}}$, and $\boldsymbol{b}^{(i)} \in \mathbb{R}^{d_{i+1}}$ denote the weights and bias of the $i$th layer; $\boldsymbol{h}^{(i)} \in \mathbb{R}^{d_i}$ marks the hidden unit at layer $i$; and $\odot$ indicates the element-wise multiplication. An intriguing aspect of the multiplicative filter network is that the final output can be expressed as a linear composition of Gabor filters:

$$
\mathcal{F}_\theta(\mathcal{U}_k, \boldsymbol{x}) = \sum_{m=1}^{M} c_k^{(m)} g\left(\boldsymbol{x}; \tau_k^{(m)}\right) + \text{bias},
\tag{5}
$$

where $M \gg L \in \mathbb{N}$, $\{c_k^{(m)}\}_{m=1}^{M}$ represents coefficients that depend on $\{\boldsymbol{W}_{\boldsymbol{h}}, \boldsymbol{W}_{\boldsymbol{z}}, \boldsymbol{b}_{\boldsymbol{h}}\}$ individually, and $\{\tau_k^{(m)}\}_{m=1}^{M}$ a set of filter parameters, i.e., $\{\boldsymbol{\gamma}, \boldsymbol{\mu}\}$. We jointly train both the encoder and decoder. Additional information about training procedures and empirical observations related to the initialization of the encoder and decoder can be found in Appendix B.2.

### 3.3 TEMPORAL ADAPTABILITY

Owing to its coordinate-based architecture, the proposed model generates a continuous representation in the spatial domain. However, replacing the conventional time index ($t$)-based INR with a context-aware indexing mechanism introduces a consideration: the direct support for continuous representation in the temporal dimension may be affected. For example: • **Reconstruction.** For time instances where $\mathcal{U}_t$ is not available. This can be accomplished through the application of interpolation techniques, such as linear or spline interpolation, to the latent codes. Subsequently, the interpolated latent code, in conjunction with the trained decoder, is employed to make inferences. • **Nowcasting.** Typically involves predicting current or very-near-future weather conditions within the next few hours, using real-time observational data. In this context, we assume access to observations for the nowcasting model. During the inference stage, the model generates a new latent code based on these observations while keeping all decoder parameters fixed. • **Forecasting.** As the decoder remains fixed after training, the extrapolation for forecasting tasks involves predicting the latent code. This can be achieved through two types of methods: autoregressive and neural operator. Autoregressive methods iteratively update the latent code $\boldsymbol{z}(t + \Delta t) = \mathcal{M}(\Delta t, \boldsymbol{z}(t))$, where $\mathcal{M} : \mathbb{R}_{>0} \times \mathbb{R}^{d_{\boldsymbol{z}}} \to \mathbb{R}^{d_{\boldsymbol{z}}}$ is the temporal update (e.g., recurrent neural network (RNN) (Sherstinsky, 2020)), while neural operator methods map a stack of latent codes to a future state/states: $\boldsymbol{z}_t = \mathcal{M}(t, \boldsymbol{z}_1, \boldsymbol{z}_2, \ldots, \boldsymbol{z}_{t-1})$ (e.g., Fourier neural operator (FNO) (Li et al., 2021)). In summary, the proposed model's temporal adaptability is contingent on the study's objectives. Detailed temporal interpolation and extrapolation can be seamlessly integrated into the proposed model as needed.

## 4 EXPERIMENTS

### 4.1 EXPERIMENTAL SETUP

**Datasets.** For these experiments, we evaluate the model's performance on two challenging datasets (additionally detailed in Appendix A): • **Simulation-based data.** The Community Earth System Model version 2 (CESM2) (Danabasoglu et al., 2020), a fully coupled global climate model, is used to simulate Earth's climate states. This work uses monthly averaged global surface temperature data, representing an atmospheric field, for model testing. Note that seasonal cycles have been removed to augment complexity, and the dataset dimensions are 1024 (time), 192 (lat), and 288 (lon). • **Satellite-based data.** Sea surface temperature data are derived from both a retrospective dataset with a four-day latency and a near-real-time dataset with a one-day latency (Martin et al., 2012). Wavelets are employed as basis functions for optimal interpolation on a global 0.01-degree grid. We analyze one year of daily data, spanning from August 20, 2022 to August 20, 2023, at the provided resolution of one-hundredth of a degree, focusing on the Gulf Stream region. The dataset dimensions are 360 (time), 901 (lat), and 1001 (lon).

**Tasks.** We assess the performance of models across diverse field reconstruction tasks to gauge their efficacy in various scenarios, including: • **Randomness.** Four tasks of increasing complexity are defined to evaluate the models' capability in handling sensor-related randomness. In the first task, models are trained with a consistent number of data points located at fixed positions. The second task introduces variability by randomly varying the number of data points. In the third task, we maintain a fixed number of data points but introduce randomness by randomly sampling grid maps, while the fourth task combines both aspects of variability, involving random sampling of the number of data points and the grid map itself. • **Sparsity.** In each task, we define three sparsity levels. Specifically, the training procedure involves using partial observations sampled from the complete state, i.e., $s \in \{5\%, 25\%, 50\%\}$ for simulation-based data and $s \in \{0.1\%, 0.3\%, 0.5\%\}$ for satellite-based data. Testing employs the complete state ($s = 100\%$). In Appendix B.1, Figure 8 provides a visual representation of the four tasks with additional task definitions available.

**Baselines.** Our model is assessed against a series of state-of-the-art implicit neural networks for field reconstruction. • **ResMLP** (Huang & Hoefler, 2023) contains a sequence of six fully connected

Table 1: Performance comparison with four INR baselines on both high-fidelity climate simulation data and real-world satellite-based benchmarks. MSE is recorded. A smaller MSE denotes superior performance. For clarity, the best results are highlighted in bold, while the second-best are underlined. The *promotion* metric, which indicates the reduction in relative error compared to the second-best model for each task, also is included.

| Model | Simulation-based Data | | | | Satellite-based Data | | | |
|---|---|---|---|---|---|---|---|---|
| | Task 1 | Task 2 | Task 3 | Task 4 | Task 1 | Task 2 | Task 3 | Task 4 |
| | Sampling ratio $s = 5\%$ | | | | Sampling ratio $s = 0.1\%$ | | | |
| ResMLP | $\underline{1.951e\text{-}2}$ | $1.672e\text{-}2$ | $1.901e\text{-}2$ | $1.468e\text{-}2$ | $\underline{1.717e\text{-}3}$ | $\underline{1.601e\text{-}3}$ | $1.179e\text{-}3$ | $1.282e\text{-}3$ |
| SIREN | $2.483e\text{-}2$ | $2.457e\text{-}2$ | $2.730e\text{-}1$ | $2.455e\text{-}2$ | $3.129e\text{-}1$ | $4.398e\text{-}2$ | $1.304e\text{-}2$ | $9.338e\text{-}2$ |
| FFN+P | $2.974e\text{-}2$ | $\underline{1.121e\text{-}2}$ | $\underline{1.495e\text{-}2}$ | $\underline{8.927e\text{-}3}$ | $2.917e\text{-}3$ | $2.392e\text{-}3$ | $\underline{7.912e\text{-}4}$ | $\underline{7.565e\text{-}4}$ |
| FFN+G | $2.943e\text{-}2$ | $1.948e\text{-}2$ | $1.980e\text{-}2$ | $1.426e\text{-}2$ | $4.904e\text{-}3$ | $7.969e\text{-}3$ | $1.005e\text{-}3$ | $1.044e\text{-}3$ |
| MMGN | $\mathbf{4.244e\text{-}3}$ | $\mathbf{4.731e\text{-}3}$ | $\mathbf{3.148e\text{-}3}$ | $\mathbf{3.927e\text{-}3}$ | $\mathbf{1.073e\text{-}3}$ | $\mathbf{1.131e\text{-}3}$ | $\mathbf{6.309e\text{-}4}$ | $\mathbf{6.298e\text{-}4}$ |
| *Promotion* | 78.24% | 57.79% | 78.94% | 56.01% | 37.51% | 29.35% | 20.26% | 16.74% |
| | Sampling ratio $s = 25\%$ | | | | Sampling ratio $s = 0.3\%$ | | | |
| ResMLP | $1.593e\text{-}2$ | $1.252e\text{-}2$ | $1.322e\text{-}2$ | $1.378e\text{-}2$ | $9.601e\text{-}4$ | $\underline{7.808e\text{-}4}$ | $8.264e\text{-}4$ | $8.144e\text{-}4$ |
| SIREN | $2.643e\text{-}2$ | $2.669e\text{-}2$ | $2.730e\text{-}1$ | $2.679e\text{-}2$ | $7.630e\text{-}3$ | $6.421e\text{-}3$ | $6.297e\text{-}3$ | $9.925e\text{-}3$ |
| FFN+P | $\underline{8.374e\text{-}3}$ | $\underline{7.905e\text{-}3}$ | $\underline{7.720e\text{-}3}$ | $\underline{6.514e\text{-}3}$ | $\underline{8.429e\text{-}4}$ | $8.294e\text{-}4$ | $\underline{4.823e\text{-}4}$ | $\underline{5.580e\text{-}4}$ |
| FFN+G | $1.307e\text{-}2$ | $1.360e\text{-}2$ | $1.331e\text{-}2$ | $1.300e\text{-}2$ | $9.169e\text{-}4$ | $1.157e\text{-}3$ | $8.128e\text{-}4$ | $6.253e\text{-}4$ |
| MMGN | $\mathbf{2.955e\text{-}3}$ | $\mathbf{2.991e\text{-}3}$ | $\mathbf{2.780e\text{-}3}$ | $\mathbf{2.802e\text{-}3}$ | $\mathbf{6.116e\text{-}4}$ | $\mathbf{5.912e\text{-}4}$ | $\mathbf{4.582e\text{-}4}$ | $\mathbf{4.896e\text{-}4}$ |
| *Promotion* | 64.71% | 62.16% | 63.98% | 56.98% | 27.44% | 24.28% | 4.99% | 12.25% |
| | Sampling ratio $s = 50\%$ | | | | Sampling ratio $s = 0.5\%$ | | | |
| ResMLP | $1.004e\text{-}2$ | $8.461e\text{-}3$ | $9.716e\text{-}3$ | $1.138e\text{-}2$ | $6.741e\text{-}4$ | $7.752e\text{-}4$ | $5.790e\text{-}4$ | $8.317e\text{-}4$ |
| SIREN | $2.728e\text{-}1$ | $2.728e\text{-}1$ | $1.408e\text{-}2$ | $1.382e\text{-}2$ | $2.214e\text{-}3$ | $9.131e\text{-}4$ | $5.973e\text{-}4$ | $8.019e\text{-}1$ |
| FFN+P | $\underline{5.383e\text{-}3}$ | $6.192e\text{-}3$ | $\underline{5.782e\text{-}3}$ | $\underline{5.503e\text{-}3}$ | $\underline{5.413e\text{-}4}$ | $\underline{5.807e\text{-}4}$ | $\underline{4.226e\text{-}4}$ | $\underline{5.004e\text{-}4}$ |
| FFN+G | $1.182e\text{-}2$ | $1.148e\text{-}2$ | $1.201e\text{-}2$ | $1.124e\text{-}2$ | $7.067e\text{-}4$ | $8.256e\text{-}4$ | $6.783e\text{-}4$ | $5.721e\text{-}4$ |
| MMGN | $\mathbf{2.802e\text{-}3}$ | $\mathbf{2.824e\text{-}3}$ | $\mathbf{2.730e\text{-}3}$ | $\mathbf{2.760e\text{-}3}$ | $\mathbf{5.081e\text{-}4}$ | $\mathbf{4.760e\text{-}4}$ | $\mathbf{4.127e\text{-}4}$ | $\mathbf{4.124e\text{-}4}$ |
| *Promotion* | 47.94% | 54.39% | 52.78% | 49.84% | 6.13% | 18.02% | 2.34% | 17.58% |

blocks, and each block consists of two fully connected feedforward layers, incorporating batch normalization and concluding with a skip connection. • **SIREN** (Sitzmann et al., 2020) is instantiated as an MLP consisting of five fully connected feedforward layers, each employing sine activation functions. • **FFN+P/G** (Tancik et al., 2020) refers to the Fourier feature network with either positional encoding (P) or Gaussian encoding (G). The network consists of a 4-layer coordinate-based MLP. It applies element-wise Fourier feature mappings to the input and uses the Gaussian error linear unit, or GELU, function for nonlinear activation. Network details are available in Appendix B.3 and B.4.

## 4.2 MAIN RESULTS

**Quantitative Evaluation.** Table 1 presents the reconstruction errors across all experiments. Notably, MMGN consistently outperforms the other baseline models. This performance advantage is particularly pronounced under low sampling ratios. Generally, as the subsampling ratio decreases, all models experience degradation in performance. MMGN still achieves significant error reductions, ranging from $56.01\%$ to $78.94\%$ for simulation-based data and $16.74\%$ to $37.51\%$ for satellite-based data when compared to the second-best model, particularly at the lowest sampling ratio. In-depth analysis of the detailed results, including extreme-case scenarios and convergence studies, can be found in Appendix C.1 and C.2.

**Qualitative Evaluation.** Figure 3 illustrates relative test errors and models' predictions. The visualization highlights the superiority of MMGN over other methods. ResMLP exhibits over-smoothed predictions, struggling to capture high-frequency signals. FFN+P displays some structural checkerboard effects, particularly noticeable in the satellite-based data. Despite quantitatively performing slightly worse than FFN+P, FFN+G demonstrates significant reconstruction improvements, successfully capturing both the overall landscape and finer details. In contrast, MMGN's prediction results faithfully reconstruct data with complex structures from sparse measurements, leading to a substantial reduction in errors. Appendix C.4 features the full results.

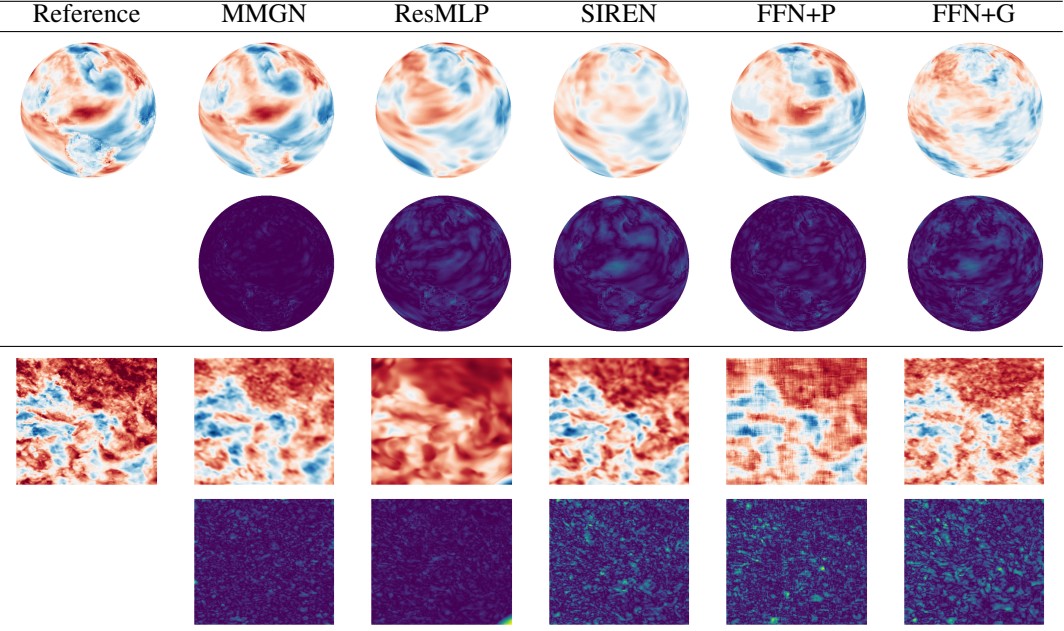

Figure 3: Visualizations of true and reconstructed fields. Global surface temperature derived from multiscale high-fidelity climate simulations and sea surface temperature assimilated using satellite imagery observations. For each dataset, the first column displays the ground truth, the first row showcases predictions from different models, and the second row presents corresponding error maps relative to the reference data. In the error maps, darker pixels indicate lower error levels.

**Robustness to Noise.** It is important to investigate the model's performance in the presence of noise. Here, we quantify the noise by the channelwise standard deviation specific to that dataset and customize noise ratios, including scenarios with noise levels set at 1%, 5%, and 10%. The analysis is conducted on the simulation dataset, and the computed results reveal that the proposed MMGN model surpasses current baselines. Notably, ResMLP does exhibit robust performance in maintaining accuracy as the noise level rises. Meanwhile, MMGN maintains its performance effectively when the noise ratio is below 5% with a noticeable increase in errors observed when the noise reaches 10%. In contrast, the performance degradation of the two FFN types is more evident.

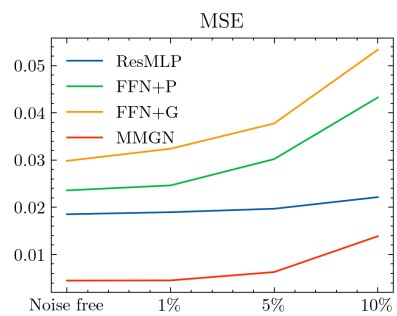

Figure 4: Model performance with different levels of noise.

**Ablations of Gabor Filter.** To assess the Gabor filter's efficacy in MMGN, we conduct detailed ablations, encompassing the removal of filter designs and their substitution with alternative types. Key observations from Table 2 include: 1) eliminating filters leads to a significant drop in model performance, highlighting the indispensability of filter designs and 2) substituting the Gabor filter with a Fourier filter not only diminishes accuracy (9.0% drop in simulation data and 18.1% decrease in satellite data), but it also increases the model size.

Table 2: Network Design Ablations. Two types of experiments were conducted: substituting the proposed Gabor filter with alternative designs and removing filters altogether. The evaluation metric is mean squared error (MSE).

| Designs | # Param | Simulation | Satellite |
|---------|---------|------------|-----------|
| None    | 577 K   | $1.758e\text{-}2$ | $6.883e\text{-}3$ |
| Fourier | 601 K   | $4.439e\text{-}3$ | $7.422e\text{-}4$ |
| Gabor   | 581 K   | $4.073e\text{-}3$ | $6.290e\text{-}4$ |

**Ablations of Context-aware Indexing Mechanism.** To evaluate the effectiveness of the proposed context-aware indexing mechanism, the latent size of MMGN is intentionally reduced to 1. In this configuration, akin to current INR baselines, the model is equipped with three inputs: $x$-coordinate, $y$-coordinate, and the latent code $z$, which is a scalar in this specific experiment. Given that $z_t$ is learned from the entirety of available measurements at time $t$, it is anticipated to encapsulate more semantic information, consequently enhancing the decoder's performance. As depicted in Figure 5, the results indicate that with a latent size reduced to 1, MMGN exhibits a slight but consistent performance improvement over the second-best model, ResMLP, across both datasets.

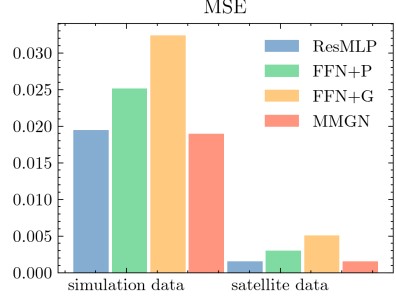

Figure 5: Ablation on context-aware indexing mechanism.

### 4.3 MODEL ANALYSIS

**Model Efficiency.** Model efficiency is evaluated based on inference speed and model size (depicted in Figure 6). Each model is executed 10 times to evaluate the entire dataset and compute the average inference speed for each instance. Overall, MMGN exhibits a favorable balance between accuracy and efficiency, making it the top-performing model. FFN+P ranks as the second-best model in most tasks (refer to Table 1), yet it possesses slightly more model parameters, resulting in reduced efficiency. Concretely, in the case of simulation data, MMGN outperforms other models with a relative speed improvement of 9.28%, 2.09%, 7.03%, and 3.65% compared to FFN+P, ResMLP, FFN+G, and SIREN, respectively. Similar results are observed in the satellite-based dataset, and additional details are provided in the Appendix C.3.

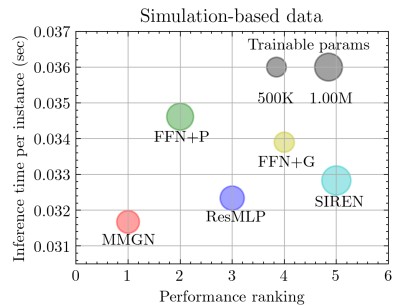

Figure 6: Efficiency Comparison. Inference time is assessed per instance.

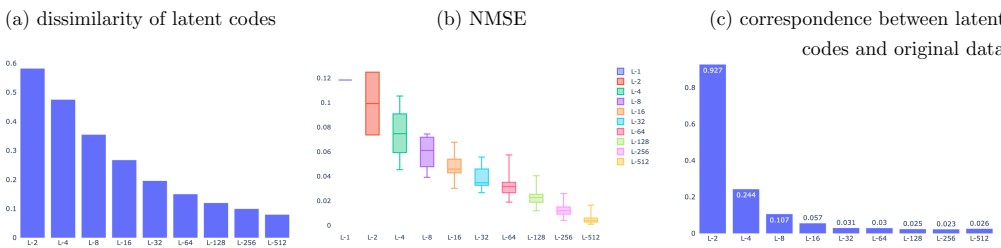

Figure 7: XAI analysis results: (a) dissimilarity between latent codes, (b) ablation of latent variable, and (c) diagnosis of auto-decoder.

**Explainable Artificial Intelligence (XAI) Analysis of Latent Codes.** To further explore the influence of the latent code $z$ on our model's performance, we train 10 models with different latent sizes, ranging from 1 to 512, by doubling the latent size at a time and collect the corresponding learned latent codes $z_{t_k}$ for all time steps $t_k$. We then assemble a matrix $Z = [z^{(1,1)}, z^{(2,1)}, z^{(2,2)}, \ldots, z^{(512,1)}, z^{(512,512)}, \ldots, z^{(512,512)}]$ with trained latent codes from different experiments. Using simulation-based data, which includes a total of 1024 time instances, results in a matrix size of 1024 by $1023 = 1 + 2 + \cdots + 512$. • **Similarity between latent codes.** For a certain latent space, we compute the pairwise Pearson Correlation of the latent codes and use the standard deviation of all the correlations to represent the overall similarity of latent variables in that latent space. In Figure 7(a), when the latent size increases, the standard deviation (dissimilarity) decreases. To further illustrate how the latent variables correlate across latent spaces, we use t-SNE (Van der Maaten & Hinton, 2008) to embed them into the same 2D plane and compare their similarities. As shown in Appendix C.5, scatter plots of the latent variables become more abundant as the latent size increases, where these latent variables are more similar. • **Ablation of latent variable.** Each variable inside the latent code is referred to as a *latent variable*. For example, if the latent code is of dimension 1024, it consists of 1024 latent variables. In our ablation study, we iteratively remove one variable at a time to regenerate the entire dataset. This process is repeated for each latent variable, and the MSE is calculated accordingly. Subsequently, we compute the percentage increase in MSE results, denoted as "NMSE." To facilitate interpretation of the ablation study across latent variables, we employ a boxplot for each latent space. The boxplot illustrates the distribution of NMSE values after ablating a specific latent variable. As depicted in the Figure 7(b), both the mean and variance of the error consistently decrease as the latent size increases. • **Diagnosis of auto-decoder.** The design of latent codes captures both temporal and spatial information through the proposed context-aware indexing mechanism. To illustrate, we consider the entire dataset and treat the latent vectors as high-dimensional temporal vectors. With 1024 temporal vectors, their dimensions vary from latent sizes (e.g., 1, 2, …, 512) to the original spatial data dimension ($192 \times 288$). For the temporal vectors in any latent space, we calculate the pairwise Pearson Correlation, comparing them with pairwise Pearson Correlation of the original temporal vectors. Figure 7(c) presents this comparison using MSE to measure the differences and showcase the correspondence of the latent vectors to the original data. We observe that as the latent dimension increases, the error consistently decreases.

## 5 CONCLUSION

This work introduces MMGN, a novel INR model for scientific data reconstruction. Compared to other time index ($t$)-based INR models, MMGN introduces a more context-aware indexing mechanism via a trainable latent code. Comprehensive experiments have been conducted to showcase the improvements facilitated by such a context-aware representation. One limitation of this current work is its focus on reconstructing a selected trajectory. In the future, we plan to investigate MMGN's potential for generalization to multiple trajectories or even across various climate properties with the goal of recovering arbitrary underlying flow maps. Concurrently, exploring the application of MMGN for optimal sensor placement is a worthwhile avenue to explore. Given the challenging combinatorial nature of optimal sensor placement, MMGN can serve as a surrogate model to expedite the search process.

ACKNOWLEDGMENTS

The authors would like to thank Klaus Tan and Avish Parmar for their efforts in analyzing the climate datasets. This material is based upon work supported by the U.S. Department of Energy (DOE)'s Office of Science, Office of Advanced Scientific Computing Research, Office of Biological and Environmental Research, and the Scientific Discovery through Advanced Computing (SciDAC) program under Award Number 9233218CNA000001. Brookhaven National Laboratory is supported by the DOE's Office of Science under Contract No. DE-SC0012704. This research used the Perlmutter supercomputer of the National Energy Research Scientific Computing Center, which is supported by DOE's Office of Science under Contract No. DE-AC02-05CH11231. The authors also thank the anonymous reviewers for their comments and suggestions that have helped to improve the manuscript's quality and clarity.

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

# A  DATASET DETAILS

For testing and validation of the proposed methodology. we use two datasets with different characteristics. While one dataset comes from a simulation of a leading climate model, the other is an observation-based dataset. Furthermore, because the ocean and atmosphere are two key dynamical components of the climate system with widely differing characteristics of spatiotemporal variability, the first dataset is an atmospheric field, while the second is an oceanic field. This choice permits us to evaluate the proposed approach's performance in reconstructing continuous fields from sparse observations using implicit neural representations (INRs) over a range of varied conditions and demonstrate its potential for a range of other applications.

## A.1  SIMULATION-BASED GLOBAL SURFACE TEMPERATURE

The first dataset is obtained from the pre-industrial control run of the Community Earth System Model version 2 (CESM2), which is a state-of-the-art climate model developed by the National Center for Atmospheric Research (NCAR). This dataset comprises monthly averaged global surface temperature data, representative of an atmospheric field. It is worth noting that the atmospheric fields are complex with spatial and temporal scales of variability that range from small-scale turbulence to large-scale weather systems and climatic patterns. The CESM2 pre-industrial control run dataset encapsulates this complexity, providing a robust testing ground for our method.

The CESM2 model simulates interactions between the various components of the earth system, including the atmosphere, oceans, land surface, and sea ice. The pre-industrial control run is designed to represent a stable climate without any anthropogenic influences, providing a baseline for

comparison against changes in future climate scenarios. The monthly averaged global surface temperature data from this run captures long-term climate variability, which is an essential factor in understanding and predicting future climate change. This dataset's rich spatial and temporal variability, combined with its simulated nature, allows us to rigorously test the performance and robustness of our proposed INR method. These data are made available under the World Climate Research Program (WCRP)'s Coupled Model Intercomparison Project (CMIP) that coordinates independent model intercomparison activities and their experiments and uses a common infrastructure for collecting, organizing, and distributing output from models performing common sets of experiments. The U.S. Department of Energy's Program for Climate Model Diagnosis and Intercomparison (PCMDI) provides coordinating support and development of software infrastructure in partnership with the Global Organization for Earth System Science Portals to provide the data at pcmdi.llnl.gov.

## A.2 SATELLITE-BASED SEA SURFACE TEMPERATURE

The second dataset considered is the Group for High Resolution Sea Surface Temperature (GHRSST) Level 4 Multiscale Ultrahigh Resolution (MUR) Global Sea Surface Temperature (SST) dataset obtained from NASA's Earthdata servers. The GHRSST Level 4 SST analysis is produced as a retrospective dataset (four-day latency) and near-real-time dataset (one-day latency) at the Jet Propulsion Laboratory (JPL) Physical Oceanography Distributed Active Archive Center (PO.DAAC) using wavelets as basis functions in an optimal interpolation approach on a global 0.01 degree grid. The version 4 MUR L4 analysis is based upon nighttime GHRSST Level 2P skin and subskin SST observations from several instruments.[1]

Selected data are from the western North Atlantic region, spanning a bounding box with the southwest corner at (34N, -69E) and the northeast corner at (44N, -59E). This oceanic dataset provides an entirely different perspective as the oceanic fields have distinct spatial and temporal scales of variability compared to atmospheric fields. Oceanic fields are characterized by a diverse range of spatial scales, from small-scale eddies and fronts to basin-wide gyres and currents. Temporally, variability can occur over short periods, such as tidal cycles, or over much longer periods, e.g., seasonal and interannual cycles. The GHRSST Level 4 MUR Global SST dataset captures this vast range of variability with the added complexity of being derived from ultra-high resolution satellite-based observations. This dataset's high resolution and comprehensiveness makes it an excellent candidate to test the effectiveness of our proposed method in managing real-world data. NASA provides full and open access to this data under its Earth Science Data Systems (ESDS) Program at earthdata.nasa.gov. One year of daily data from Aug. 20, 2022 to Aug. 20, 2023 is used at the provided resolution of a hundredth of a degree in the region of Gulf Stream (as previously specified).

## B EXPERIMENT DETAILS

### B.1 TASKS

We establish a range of tasks with increasing complexity to primarily assess the models' performance in handling the *randomness* and *sparsity* inherent in the observation sensors. In tasks where the quantity of available sensors varies randomly across different time instances, this variation is achieved by sampling a subset of data points from a predefined grid. For example, with a $5\%$ sampling rate, we initially create a grid map with $5\%$ of the data points then randomly sample from a uniform distribution within the range of $\mathcal{U}[1, 5]$, resulting in different grid maps. Figure 8 provides a visual representation of the four tasks for evaluating model performance with increasing complexity.

---

[1]They include the NASA Advanced Microwave Scanning Radiometer-EOS (AMSR-E), JAXA Advanced Microwave Scanning Radiometer 2 on GCOM-W1, Moderate Resolution Imaging Spectroradiometers (MODIS) on the NASA Aqua and Terra platforms, U.S. Navy microwave WindSat radiometer, Advanced Very High Resolution Radiometer (AVHRR) on several National Oceanic and Atmospheric Administration (NOAA) satellites, and *in situ* SST observations from the NOAA iQuam project. Ice concentration data are from the archives at the EUMETSAT Ocean and Sea Ice Satellite Application Facility (OSI SAF) High Latitude Processing Center and are also used for an improved SST parameterization for the high latitudes.

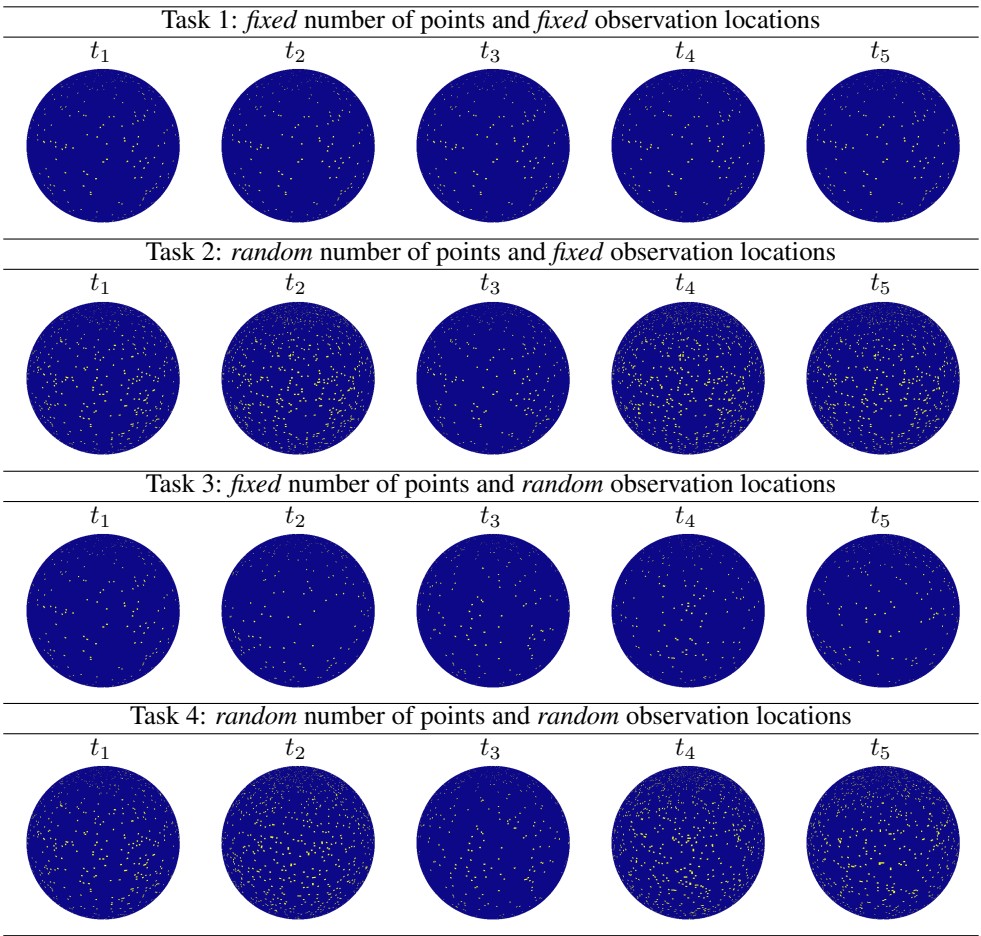

Figure 8: Four field reconstruction tasks with increasing complexity for model evaluation.

## B.2 MODEL TRAINING

Given the previously defined *auto-decoder* and *Gabor filter*-based network, we assume that the model's outputs adhere to a Gaussian distribution:

$$\boldsymbol{u}(t_k, \boldsymbol{x}) \sim \mathcal{N}\left(\mathcal{F}_\theta(\mathcal{U}_k, \boldsymbol{x}), \sigma_u^2 I\right), \tag{6}$$

where $\sigma_u^2$ is noise variance and $I$ is identity matrix. We assume the likelihood in the posterior calculation (Eq. **??**) follows the form of $p_\theta\left(u^{(i)} \mid \boldsymbol{z}_k; \boldsymbol{x}^{(i)}\right) = \exp\left(-\mathcal{L}\left(\mathcal{F}_\theta(\mathcal{U}_k, \boldsymbol{x}), u^{(i)}\right)\right)$. The network prediction, represented as $\hat{u}_k^{(i)}$, is associated with its own latent code $\boldsymbol{z}_k$. The loss function $\mathcal{L}(\hat{u}_k^{(i)}, u_k^{(i)})$ penalizes deviations between the network prediction and the actual observation. During training, we maximize the joint log posterior over all training instances $\mathcal{T} = \{t_k \in \mathbb{R}_+\}_{k=1}^M$:

$$\underset{\theta, \{\boldsymbol{z}_k\}_{k=1}^M}{\arg\min} \sum_{k=1}^M \left(\sum_{i=1}^N \mathcal{L}\left(\mathcal{F}_\theta\left(\boldsymbol{z}_k, \boldsymbol{x}^{(i)}\right), u^{(i)}\right) + \frac{1}{\sigma_u^2}\|\boldsymbol{z}_k\|_2^2\right). \tag{7}$$

Each training instance is initialized with its respective latent code $\boldsymbol{z}_k$. To optimize a specific latent code $\boldsymbol{z}_k$, we first map it to network parameters, calculate the reconstruction loss, and backpropagate this loss to $\boldsymbol{z}_k$ and other model parameters for a gradient descent update. This gradient descent update is illustrated by the red and yellow lines in Figure 2. During training, this optimization is performed jointly. During inference, the network parameters are fixed, and our attention is solely

on optimizing the latent code $z^\star$ to minimize the reconstruction error. Specifically, auto-decoders employ first-order optimization to solve $\arg\max p(z^\star|u^{(\star,1)}, u^{(\star,2)}, \dots)$, effectively using the optimization process as an encoder that maps the new observation set $\mathcal{U}^\star$ to latent variables $z^\star$. This is equivalent to evaluating $z^\star$ using Maximum-a-Posterior (MAP) estimation:

$$z^\star = \arg\min_{z^\star} \sum_{\left(x^{(\star,i)}, u^{(\star,i)}\right) \in \mathcal{U}^\star} \mathcal{L}\left(\mathcal{F}_\theta\left(z^\star, x^{(\star,i)}\right), u^{(\star,i)}\right) + \frac{1}{\sigma_u^2}\|z^\star\|_2^2. \tag{8}$$

**Decoder initialization.** Our experiments reveal that a crucial factor to consider is the initialization of the $\gamma$ and $\mu$ parameters. Given that $\gamma$ effectively functions as an inverse covariance term for a Gaussian distribution, a reasonable choice is to use a Gamma random variable $\gamma \sim \Gamma(\alpha, \beta)$, which is the conjugate prior of the Gaussian inverse covariance. On the other hand, we uniformly distribute $\mu$ over the allowable input space range $\mu \sim \mathcal{U}(x_{min}, x_{max})$.

### B.3 MODEL ARCHITECTURE AND HYPERPARAMETERS

Designing network architectures and optimizing hyperparameters pose significant challenges in deep learning, often relying on empirical and problem-specific approaches. To ensure a fair comparison, we conduct thorough hyperparameter tuning and architecture search for each model. Table 3 provides detailed model configurations for ResMLP, SIREN, FFN+P/G, and MMGN (Multiplicative and Modulated Gabor Network).

Table 3: Model configurations, hyperparameters, and associated ranges.

| Model | Hyperparameters | Values |
|---|---|---|
| ResMLP | Width | $\{128, 160, 192, 224, 256, 288, 320, 352, 384, 416, 448, 480, 512\}$ |
| | Depth | $\{3, 4, 5, 6, 7\}$ |
| | Non-linear activation | { ReLU, Sigmoid, Tanh, SELU, GELU, Swish } |
| SIREN | Width | $\{128, 192, 256, 384, 512\}$ |
| | Depth | $\{3, 4, 5, 6, 7\}$ |
| | Weight scale $w_0$ | $\{1, 5, 10, 15, 20, 25, 30, 35, 40, 45, 50, 100\}$ |
| FFN+P | Width | $\{128, 192, 256, 384, 512\}$ |
| | Depth | $\{3, 4, 5, 6, 7\}$ |
| | Frequency constant | $\{30, 40, 50, 60, 70, 80, 90, 100, 110, 120\}$ |
| | Frequency number | $\{150, 160, 170, 180, 190, 200, 210, 220, 230, 240, 250\}$ |
| | Non-linear activation | { ReLU, Sigmoid, Tanh, SELU, GELU, Swish } |
| FFN+G | Width | $\{128, 192, 256, 384, 512\}$ |
| | Depth | $\{3, 4, 5, 6, 7\}$ |
| | Gaussian $\sigma$ | $\{1, 3, 5, 7, 10, 20, 30, 40, 50\}$ |
| | Encode size | $\{64, 128, 256, 512\}$ |
| | Non-linear activation | { ReLU, Sigmoid, Tanh, SELU, GELU, Swish } |
| MMGN | Width | $\{128, 192, 256, 384, 512\}$ |
| | Depth | $\{3, 4, 5, 6, 7\}$ |
| | Input scale | $\{128, 256, 512\}$ |
| | Latent size | $\{1, 2, 4, 8, 16, 32, 64, 128, 256, 512\}$ |
| | Latent initialization | { Uniform, Gaussian, Ones, Zeros, Orthogonal } |

### B.4 BASELINE IMPLEMENTATION

All models are trained with an L2 loss function for 200 epochs, employing the AdamW optimizer (Loshchilov & Hutter, 2019) with an initial learning rate of 0.001. We apply a learning rate decay of 0.99 for each parameter group of every epoch. Due to variations in resolution between earth simulation and satellite imagery data, we set batch sizes to 16 and 2, respectively. We conduct all experiments using PyTorch Lightning, repeating each experiment 10 times on a single NVIDIA A100 40 GB GPU.

## C ADDITIONAL RESULTS

### C.1 EXTREME SPARSITY

The aim of this set of experiments is to assess how different models perform when dealing with extremely limited spatial sensor coverage, typically less than $1\%$. This situation is common in various engineering and scientific applications. For instance, oceanographic buoys and research vessels may have sparse spatial distribution in vast oceans, sometimes being hundreds of kilometers apart. Here, the focus is on satellite-based data. Quantitatively, the proposed MMGN model outperforms other INR models when the sensor coverage is as low as $0.1\%$, achieving at least a $16.74\%$ improvement. This improved performance is evident in the visual representations presented in Figure 10. Notably, models like SIREN and FFN+P struggle to produce meaningful results in such sparse scenarios. While quantitatively less accurate than FFN+P, FFN+G exhibits better qualitative predictions, particularly in Tasks 3 and 4. Interestingly, ResMLP appears to provide more realistic predictions than models based on Fourier features. However, it falls short in capturing high-frequency signals within the climate data and is sensitive to local irregularities, leading to substantial errors as observed in Task 4. In comparison to these baseline models, our proposed MMGN consistently delivers reliable reconstruction results even when working with extremely sparse data.

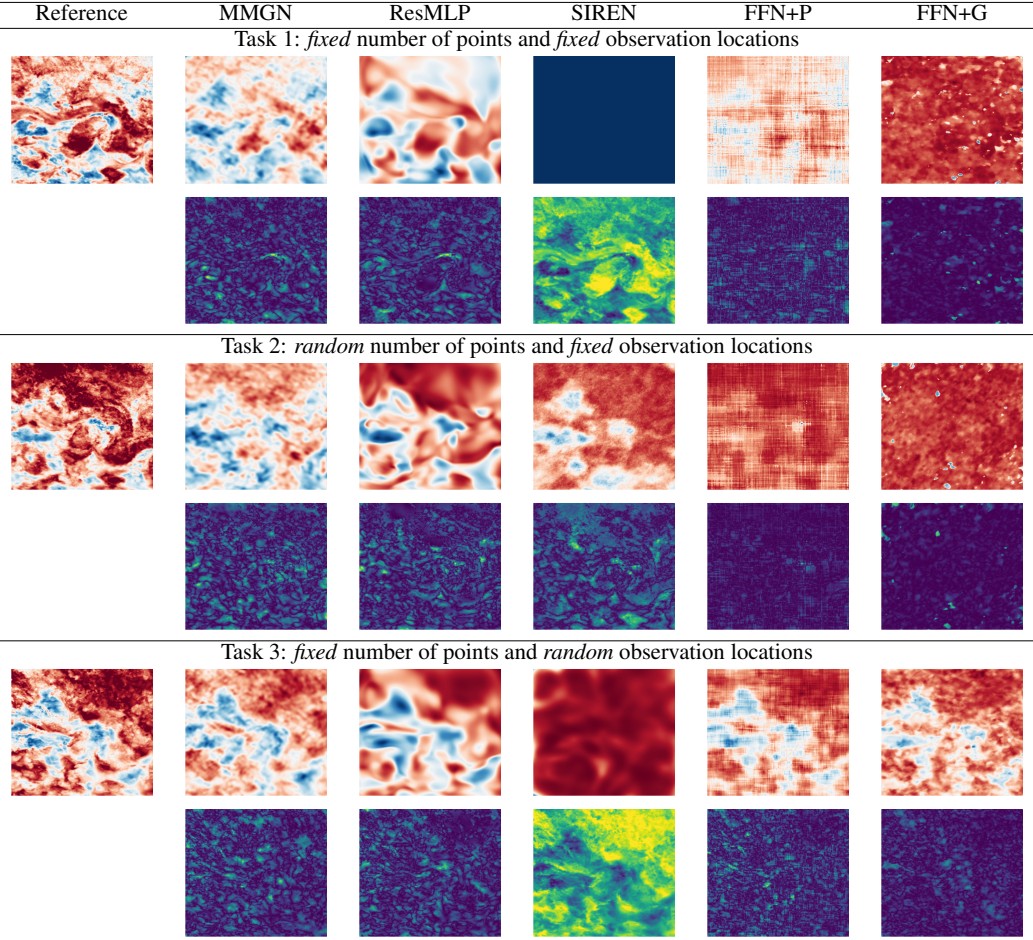

Figure 9: Comparison of reconstruction performance on ultra-high resolution satellite-based sea surface temperature data with sampling ratio $s = 0.1\%$. The first, third, and fifth rows depict model predictions, while the second, fourth, and sixth rows illustrate the errors quantified by absolute error. Darker pixels correspond to lower error values.

| Reference | MMGN | ResMLP | SIREN | FFN+P | FFN+G |
|-----------|------|--------|-------|-------|-------|

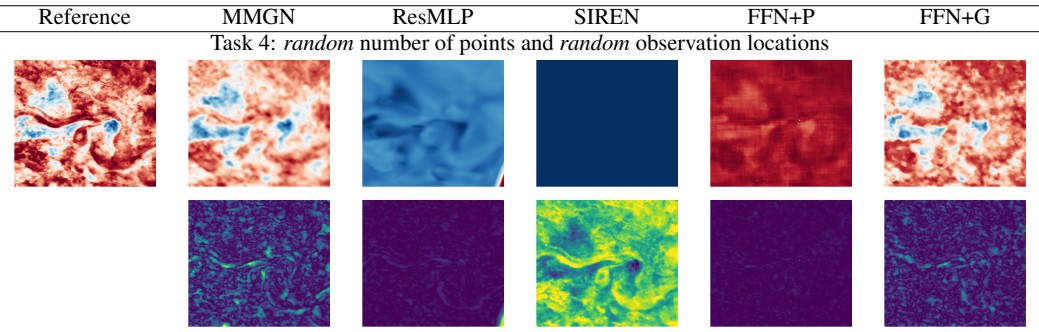

Task 4: *random* number of points and *random* observation locations

To further explore MMGN's ability to manage extremely sparse data, we conduct additional experiments with sampling ratios of 0.01%, 0.03%, 0.05%, 0.07%, and 0.09%. In addition to the mean squared error (MSE), we also compute the Peak Signal-to-Noise Ratio (PSNR) and the Structural Similarity Index (SSIM) to assess performance. The results, illustrated in Figure 10, reveal a substantial decline in performance when transitioning from 0.03% to 0.01% sampling ratios. However, as the sampling ratio increases from 0.03% to 0.09%, there is nearly linear improvement in performance. Therefore, to achieve a reliable reconstruction, especially for satellite-based data, a minimum of 0.03% data coverage is needed.

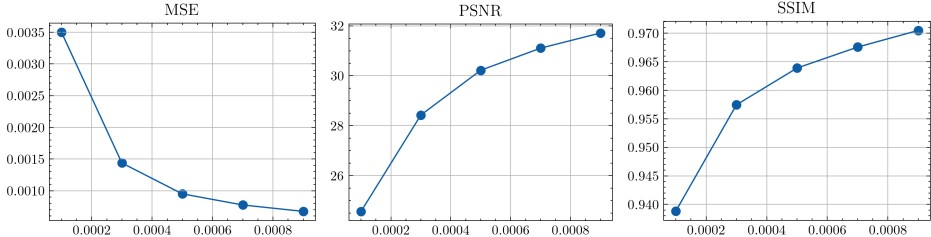

Figure 10: Impact of sampling ratios on MMGN performance in extreme sparse data scenarios.

## C.2 CONVERGENCE STUDY

To determine the minimum number of sensors required for effective reconstruction, we conduct a convergence study. The exhaustive grid search of sampling rates can be computationally demanding. To expedite this process and establish a connection with classical reconstruction methods, we initially use proper orthogonal decomposition (POD). Specifically, for each dataset, we employ POD and sort the computed eigenvalues in descending order. Once cumulative energy is calculated, approximately 750 POD modes are needed for the simulation-based data to reconstruct 90% of the total

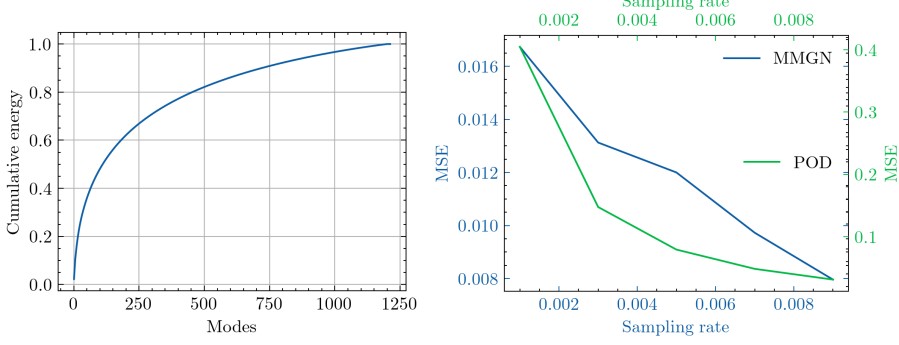

Figure 11: Cumulative energy (left) and reconstruction performance (right).

kinetic energy (Figure 11). Notably, this calculation is based on the snapshot POD method (Taira et al., 2017; Sirovich, 1987). Using this information, we define an interval $[0.1\%, 1\%]$ and conduct a grid search to estimate the minimal sampling rate. Both quantitative and qualitative findings indicate that $1\%$ is the closest approximation to the minimum sampling rate for the simulation data ( Figure 12).

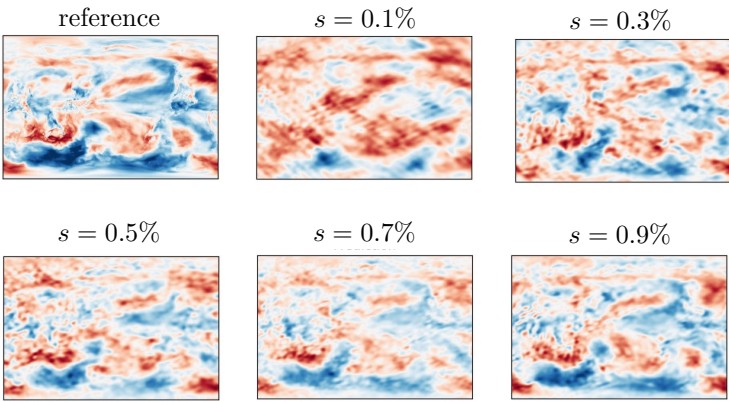

Figure 12: Convergence of sampling in MMGN.

## C.3 MODEL EFFICIENCY

Similar to Figure 6, we assess model efficiency by considering both inference speed and model size. Inference time is evaluated across all available data points for each instance with each model executed 10 times to unbiasedly evaluate the entire dataset. The resulting average inference speed for each instance then is computed. Notably, MMGN strikes an advantageous balance between accuracy and efficiency, emerging as the top-performing model in these evaluations. While FFN+G and SIREN exhibit faster inference speeds, they falter in delivering reliable results. In contrast, ResMLP stands out as the most computationally intensive model to run. While slightly faster than MMGN, FFN+P lags behind in terms of accuracy.

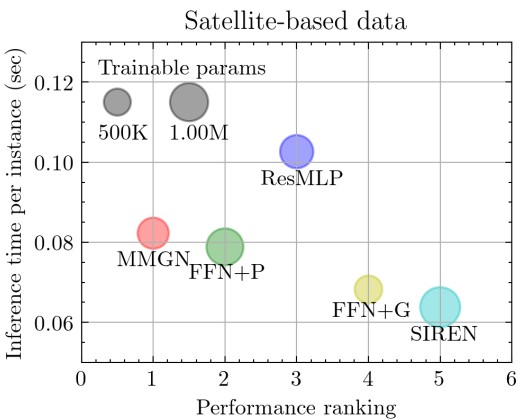

Figure 13: Efficiency Comparison. Inference time is assessed across all available data points for each instance.

## C.4 FULL QUALITATIVE RESULTS

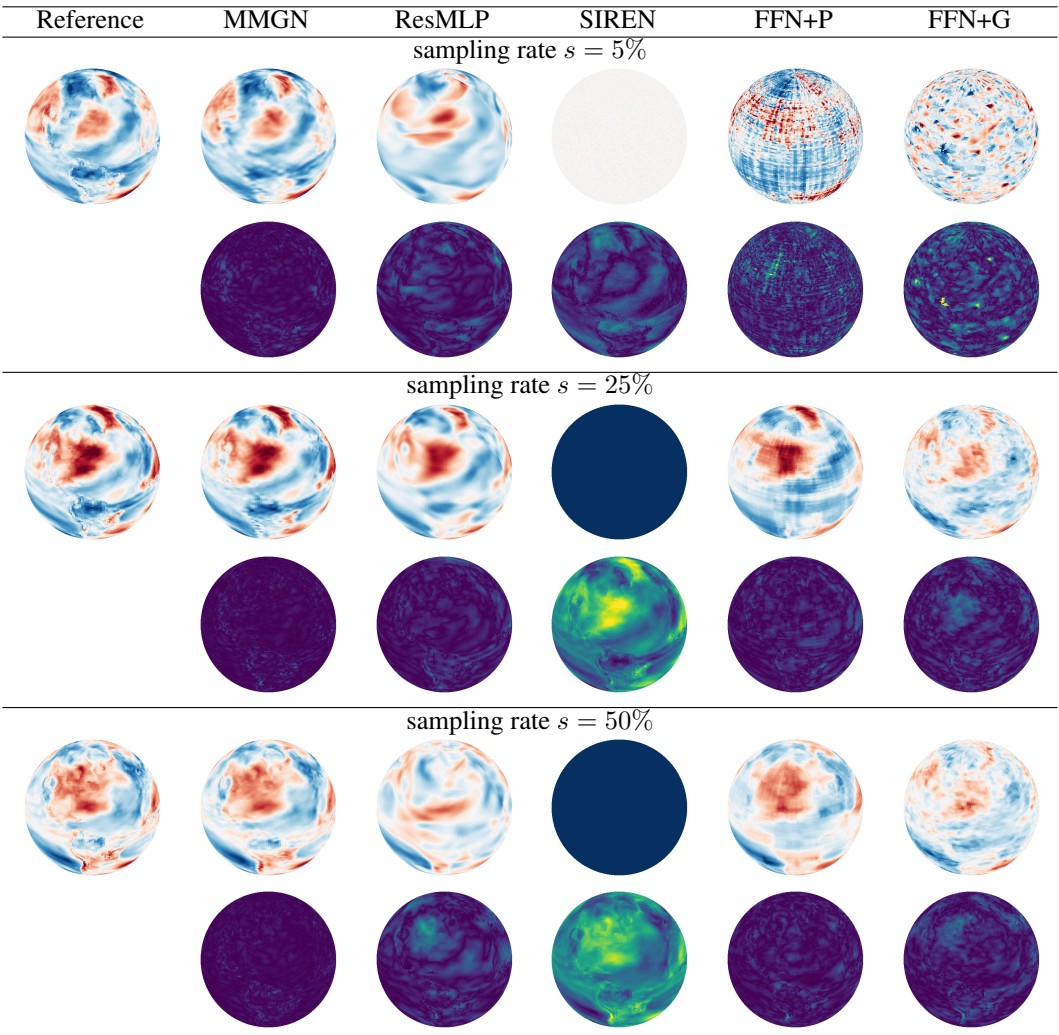

Figure 14: Comparison of reconstruction performance for Task 1 on simulation data from a state-of-the-art climate model. The first, third, and fifth rows depict model predictions, while the second, fourth, and sixth rows illustrate the errors quantified by absolute error. Darker pixels correspond to lower error values.

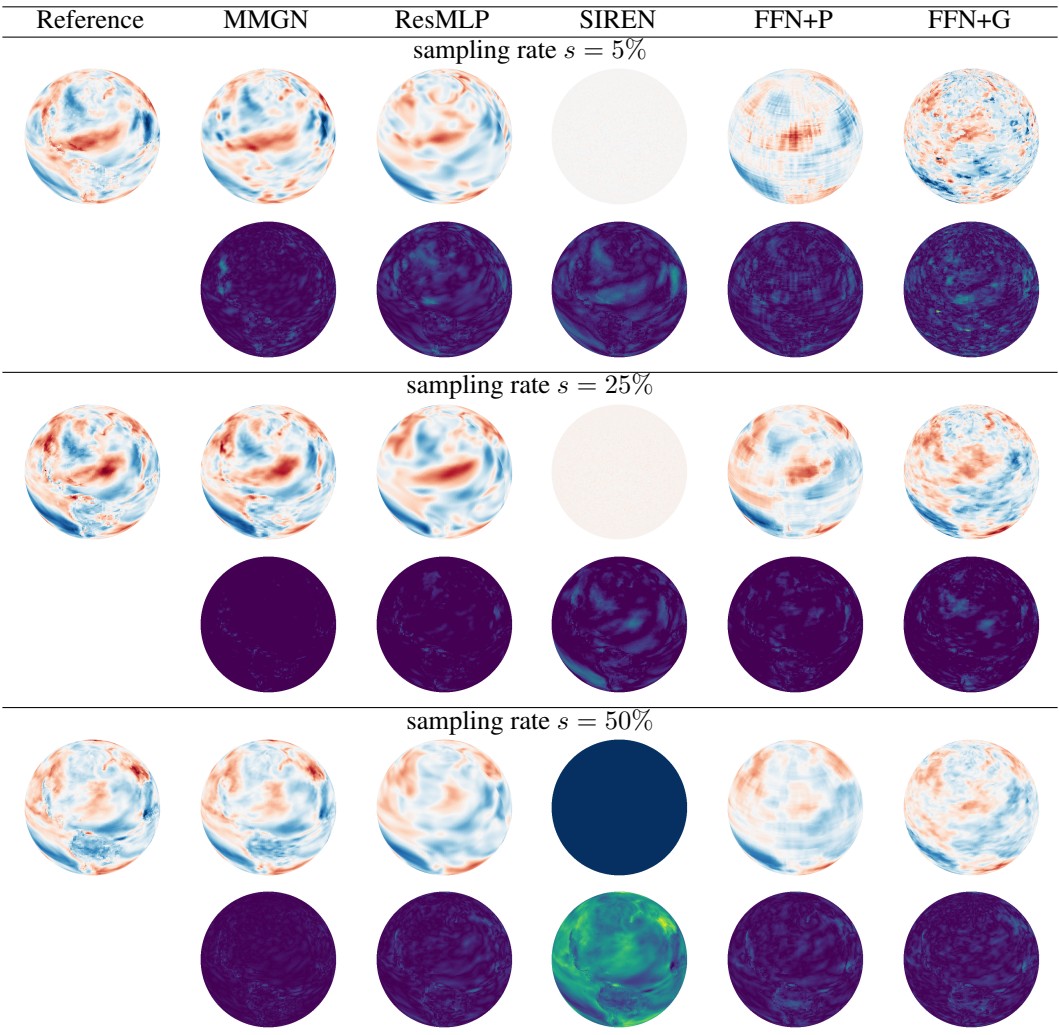

Figure 15: Comparison of reconstruction performance for Task 2 on simulation data from a state-of-the-art climate model. The first, third, and fifth rows depict model predictions, while the second, fourth, and sixth rows illustrate the errors quantified by absolute error. Darker pixels correspond to lower error values.

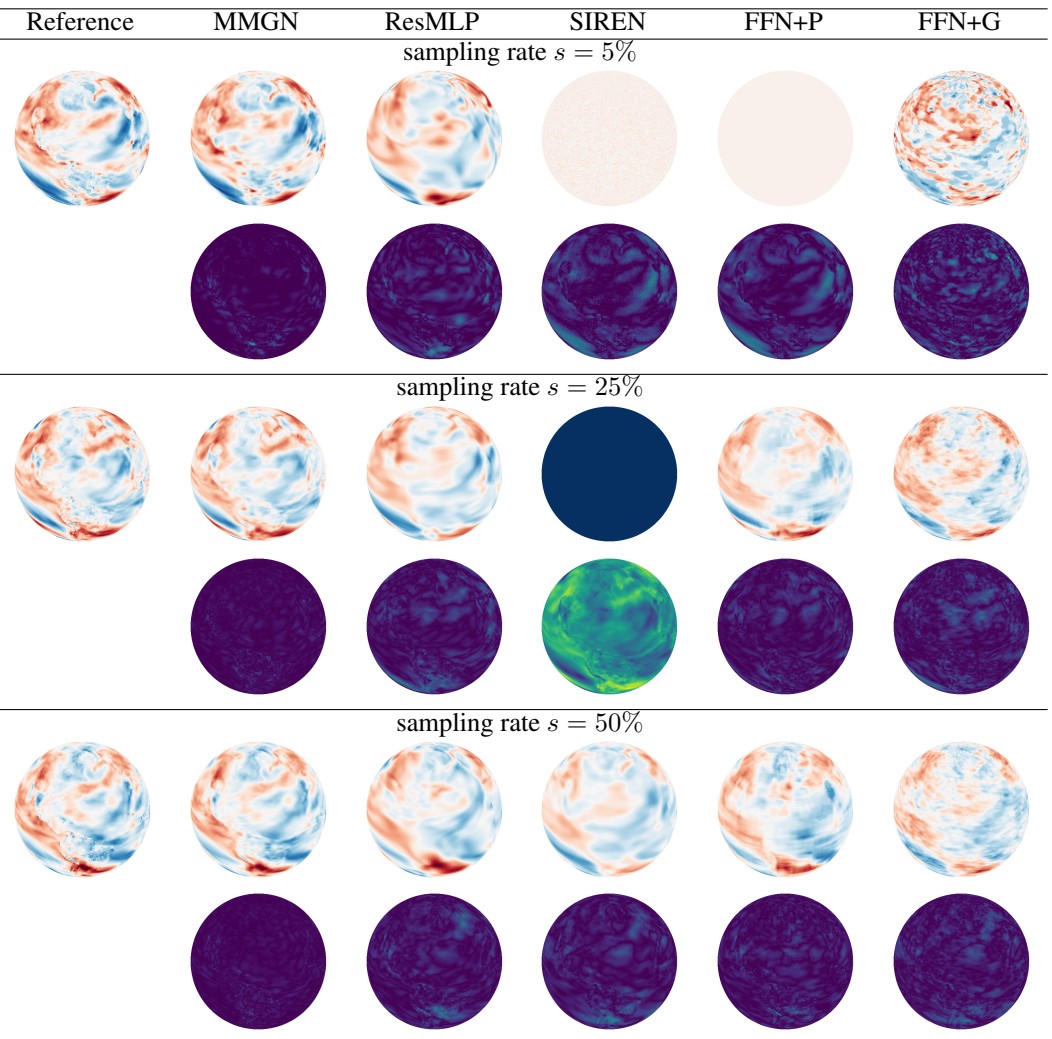

Figure 16: Comparison of reconstruction performance for Task 3 on simulation data from a state-of-the-art climate model. The first, third, and fifth rows depict model predictions, while the second, fourth, and sixth rows illustrate the errors quantified by absolute error. Darker pixels correspond to lower error values.

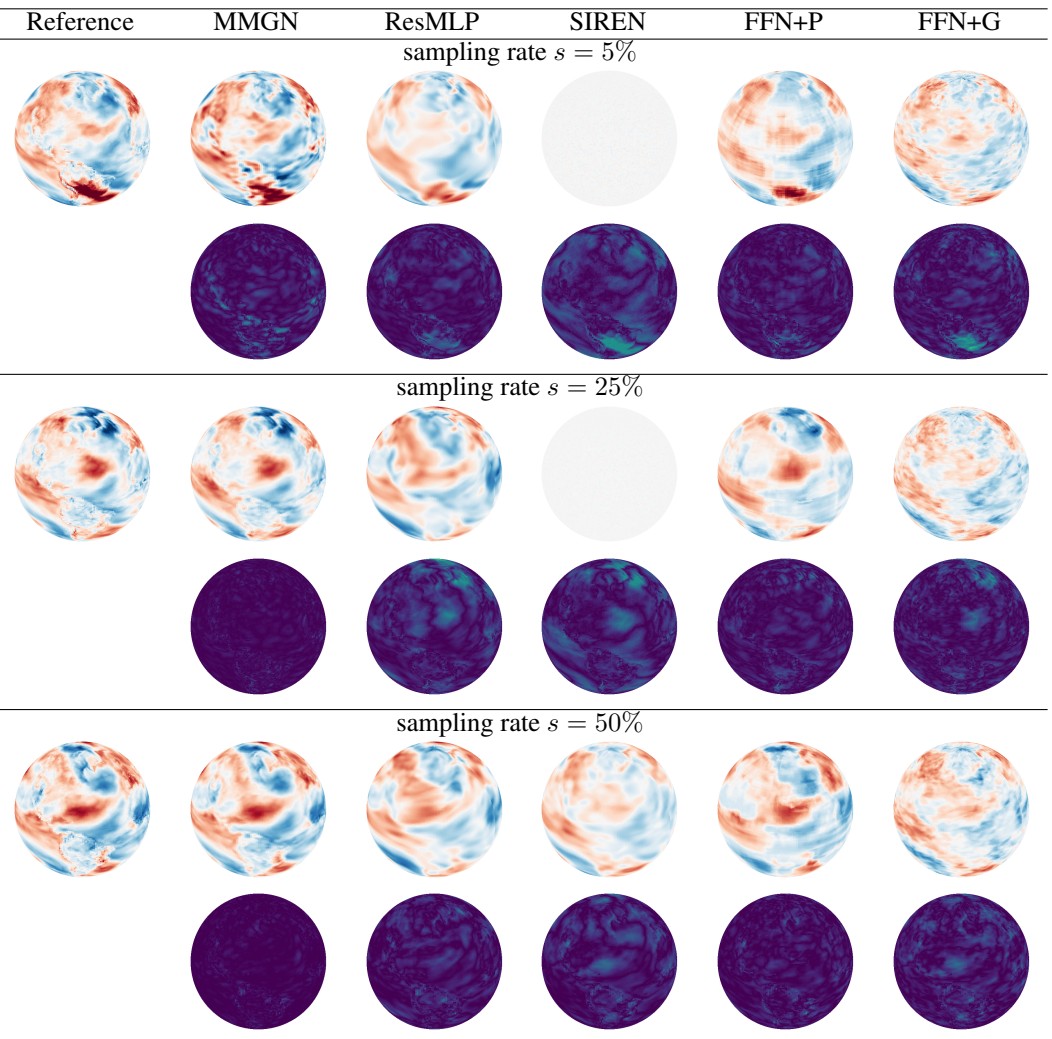

Figure 17: Comparison of reconstruction performance for Task 4 on simulation data from a state-of-the-art climate model. The first, third, and fifth rows depict model predictions, while the second, fourth, and sixth rows illustrate the errors quantified by absolute error. Darker pixels correspond to lower error values.

C.5 MODEL EXPLANATION

Figure 18 shows comprehensive results regarding the influence of latent size by comparing the similarity of latent variables across all models. Note the experiments are conducted using the simulation-based global surface temperature data with a sampling ratio of $s = 5\%$ for demonstration. We trained 10 models with varying latent sizes, specifically, $d_z \in \{1, 2, 4, 8, 16, 32, 64, 128, 256, 512\}$, while keeping the number of time steps constant at $t_k = 1024$. For instance, when utilizing a model with a latent size of 32, it operates on a 32-dimensional latent vector corresponding to the 1024 time steps. Thus, a matrix of latents for 10 models is formed, where a column (of size $M \times 1$) represents a dimension of latent vectors throughout all time steps (we call it a *latent variable*), and a row (of size $1 \times 1023$) concatenates the latents of all models in a time step (we call them *latent vectors*). Concretely, the assembled matrix can be represented as:

$$Z = [\boldsymbol{z}^{(1,1)}, \boldsymbol{z}^{(2,1)}, \boldsymbol{z}^{(2,2)}, \dots, \boldsymbol{z}^{(512,1)}, \boldsymbol{z}^{(512,2)}, \dots, \boldsymbol{z}^{(512,512)}]. \qquad (9)$$

Next, we use t-SNE to reduce the dimension of all the latent variables $Z$ and project them into a two-diemsional (2D) space to understand the similarity of latent variables. Figure 18 illustrates the distribution changes of all latent variables. Compared with the distribution of latent size 128, the cases with lower latent sizes (e.g., 1, 2, 4, and 8) show sparse sampling at nearby variables. Their latent spaces simply are the subspaces of the latent space spanned by 128 variables. When the size increases, the sampling keeps improving until the case with the size of 512 overly samples the space. Hence, we find the necessity of increasing the latent size to an appropriate granularity. However, the question arises: *What constitutes an ideal latent size*? The final row of the figure serves as a useful illustration in this regard. We begin to observe overlap between latent sizes of 64 and 128. This overlap becomes more pronounced at latent size 256. Consequently, while increasing the latent size may enhance overall coverage, it also leads to information redundancy and overlapping areas.

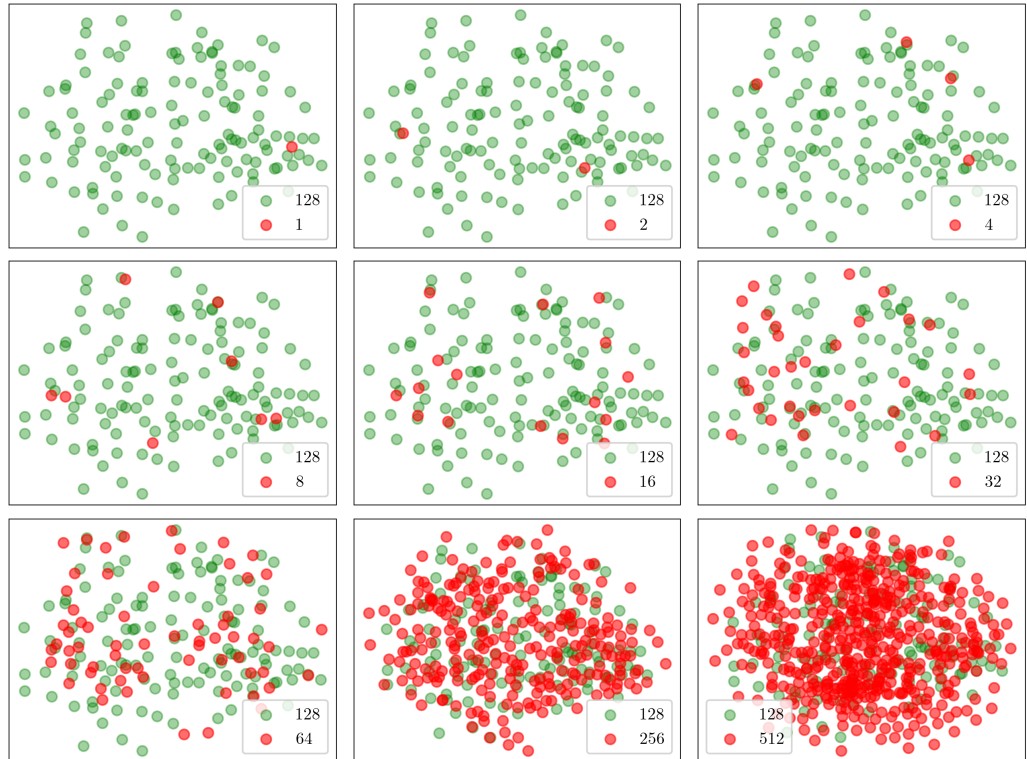

Figure 18: The impact of latent size by comparing the similarity of all the models' latent variables.

To additionally demonstrate the influence of latent variables, we systematically ablate one latent variable at a time to regenerate the entire data and assess the performance quantitatively using vari-

ous error metrics (i.e., MSE, PSNR, and SSIM). In Figure 19, Figure 20, and Figure 21, the results of all cases of latent sizes are concatenated along the x-axis, where the x-coordinate is the index of the ablated latent variable. To facilitate the comparison, we show the original error metrics without ablation with the prefix "o" before the error metric names. Observations indicate that larger latent sizes enhance reconstruction quality, individual contributions diminish as latent sizes increase, and differences in contribution exist among latent variables within the same space but are less pronounced compared to disparity across latent sizes.

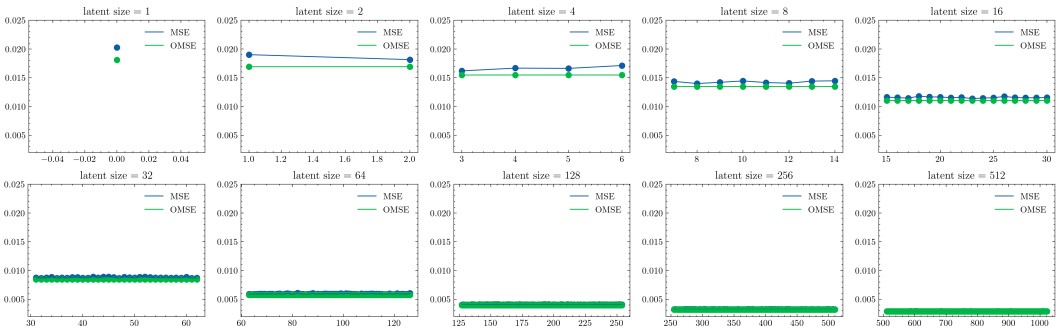

Figure 19: The ablation study of latent variables with MSE and OMSE.

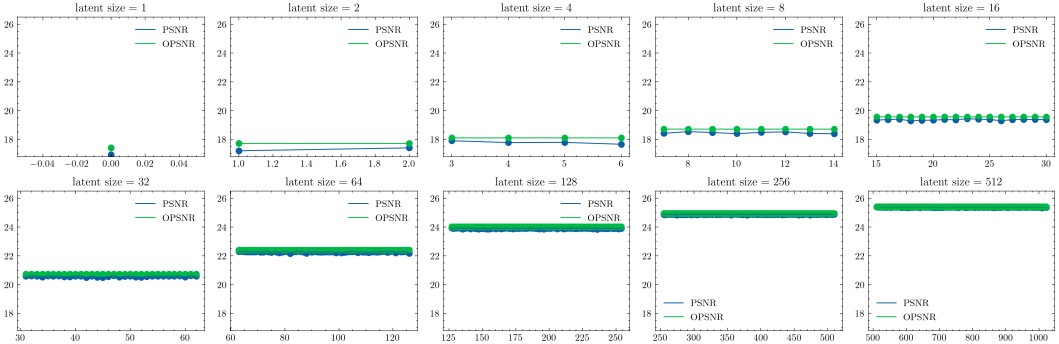

Figure 20: The ablation study of latent variables with PSNR and OPSNR.

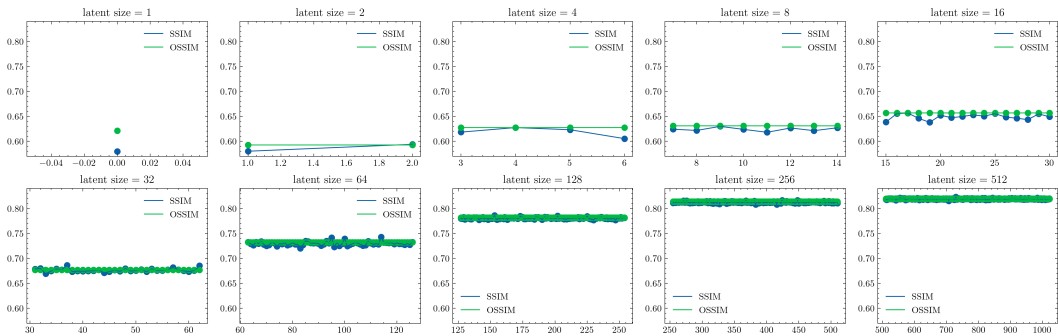

Figure 21: The ablation study of latent variables with SSIM and OSSIM.

