# OpenReview forum: "Continuous Field Reconstruction from Sparse Observations with Implicit Neural Networks"
_ICLR.cc/2024/Conference — ICLR 2024 poster_

### Official Review · Reviewer_xFQb · 2023-10-23

**Soundness:** 3 good
**Presentation:** 3 good
**Contribution:** 3 good
**Rating:** 6
**Confidence:** 4

**Summary:**

This paper proposes a new method to reconstruct spatiotemporal dynamics from sparse measurements by considering implicit neural networks. The spatiotemporal dynamics are separated into two independent spatial and temporal components. Superior performance has been shown in this paper compared with baseline models.

**Strengths:**

- This paper proposes a new pipeline by combining implicit neural networks and functional separation of variables. In addition, Gabor filters are applied for learning high-frequency signals.
- This paper is well-written and easy to follow.

**Weaknesses:**

- The dataset is not that challenging. This paper considers monthly-averaged temperature data in **Simulation-based data** CESM2. The dynamics are relatively smoother than daily-averaged data. Please see some of my follow-up concerns regarding the dataset in **Questions**.

- The sparsity setting considers some random sampling schemes. If the authors can discuss a bit on the optimal sensor placement, it would enhance the paper.

- Since the authors use the Gabor filter instead of other common schemes to learn high-frequency signals, it would be good to have an ablation study on the Gabor filter part.

**Questions:**

- The 4th line of Page 6, “we uniformly distributing” should be “we uniformly distribute”.

- How do the authors determine the sparse setting (Page 7) for those two datasets? The sparsity ratios for those two datasets are quite different. Also, can the authors provide an estimate or convergence analysis of the smallest number of sensors needed for reconstruction?

- Why not consider the ERA5 dataset for climate modeling since the recent climate modeling papers [1-3] all use this benchmark dataset?

- What are the spatial resolutions for the datasets used in this paper?

---

Refs:

[1] Bi, Kaifeng, et al. "Pangu-weather: A 3d high-resolution model for fast and accurate global weather forecast." arXiv preprint arXiv:2211.02556 (2022).

[2] Pathak, Jaideep, et al. "Fourcastnet: A global data-driven high-resolution weather model using adaptive fourier neural operators." arXiv preprint arXiv:2202.11214 (2022).

[3] Lam, Remi, et al. "GraphCast: Learning skillful medium-range global weather forecasting." arXiv preprint arXiv:2212.12794 (2022).

---

> ### Author Response · Authors · 2023-11-17
> **Response to Reviewer xFQb**
>
> We are grateful for the reviewer's constructive feedback, particularly the recommendations regarding (1) additional ablation experiments on network design; (2) optimal sensor placement; and (3) explanation of sampling rates. In the following, we address all the comments individually for clarity and completeness.
>
> ---
>
> Comment $\#$ 1: The 4th line of Page 6, “we uniformly distributing” should be “we uniformly distribute”.
>
> Response: Thank you for pointing it out. Fixed.
>
> ---
>
> Comment $\#$ 2: How do the authors determine the sparse setting (Page 7) for those two datasets? The sparsity ratios for those two datasets are quite different. Also, can the authors provide an estimate or convergence analysis of the smallest number of sensors needed for reconstruction?
>
> Response: We have conducted additional analysis to determine the minimum number of sensors required for effective reconstruction. Specifically, for each dataset, we employed proper orthogonal decomposition (POD) and sorted the computed eigenvalues in descending order. The cumulative energy was calculated, and for the simulation-based data, approximately 750 POD modes are needed to reconstruct $90\%$ of the total kinetic energy. Note this calculation is based on the snapshot POD method [1, 2]. Using this information, we defined an interval $[0.1\%, 1\%]$ and conducted a grid search to estimate the minimal sampling rate. Our findings, both quantitative and qualitative, indicate that $1\%$ is the closest approximation to the minimum sampling rate for the simulation data. Below is the MSE comparison between MMGN and POD reconstruction, with detailed results and descriptions included in the revised paper.
>
> | Sampling rate | POD | MMGN |
> | ----------- | ----------- | ----------- |
> |  $0.1 \%$ | $0.40475$ | $0.01672$ |
> |  $0.3 \%$ | $0.14760$ | $0.01312$ |
> |  $0.5 \%$ | $0.07922$ | $0.01199$ |
> |  $0.7 \%$ | $0.04830$ | $0.00972$ |
> |  $0.9 \%$ | $0.03106$ | $0.00796$ |
>
>
> [1] Taira, K., Brunton, S.L., Dawson, S.T., Rowley, C.W., Colonius, T., McKeon, B.J., Schmidt, O.T., Gordeyev, S., Theofilis, V. and Ukeiley, L.S., 2017. Modal analysis of fluid flows: An overview. Aiaa Journal, 55(12), pp.4013-4041.
>
> [2] Sirovich, L., 1987. Turbulence and the dynamics of coherent structures. I. Coherent structures. Quarterly of applied mathematics, 45(3), pp.561-571.
>
> ---
>
> Comment $\#$ 3: Why not consider the ERA5 dataset for climate modeling since the recent climate modeling papers [1-3] all use this benchmark dataset?
>
> Response: The model is designed to be data-agnostic, making it applicable to the ERA5 dataset. We plan to release the implementations as open source, enabling readers to reproduce the presented results and apply the model to datasets of their research interest.
>
> ---
>
> Comment $\#$ 4: What are the spatial resolutions for the datasets used in this paper?
>
> Response: Thank you for bringing this to our attention. For the simulation-based data, it's 1024 (time) $\times$ 192 (latitude) $\times$ 288 (longitude); and for the satellite-based data, it's 360 (time) $\times$ 901 (latitude) $\times$ 1001 (longitude). We have also accordingly revised the data description section in the paper.
>
> ---
>
> Comment $\#$ 5: The dataset is not that challenging. This paper considers monthly-averaged temperature data in Simulation-based data CESM2. The dynamics are relatively smoother than daily-averaged data. Please see some of my follow-up concerns regarding the dataset in Questions.
>
> Response: Thank you for pointing it out. In the data preprocessing step, we intentionally remove the temporal mean and seasonal cycle as they are straightforward to predict. This clarification has been included in the revised paper.

---

> > ### Author Response · Authors · 2023-11-17
> > **Response to Reviewer xFQb**
> >
> > Comment $\#$ 6: The sparsity setting considers some random sampling schemes. If the authors can discuss a bit on the optimal sensor placement, it would enhance the paper.
> >
> > Response: Thank you for the excelelnt suggetion. We included some discussion in the revised paper. In utilizing MMGN for optimal sensor placement, two approaches can be considered. Firstly, employing state-of-the-art methods such as POD-QDEIM [1,2] to identify optimal sensor locations and then utilizing MMGN trained with measurements from these identified locations for field reconstruction. Secondly, given the challenging combinatorial nature of optimal sensor placement, MMGN can serve as a surrogate model to expedite the search process.
> >
> > [1] Drmac, Z. and Gugercin, S., 2016. A new selection operator for the discrete empirical interpolation method---improved a priori error bound and extensions. SIAM Journal on Scientific Computing, 38(2), pp.A631-A648.
> >
> > [2] Manohar, K., Brunton, B.W., Kutz, J.N. and Brunton, S.L., 2018. Data-driven sparse sensor placement for reconstruction: Demonstrating the benefits of exploiting known patterns. IEEE Control Systems Magazine, 38(3), pp.63-86.
> >
> > ---
> >
> > Comment $\#$ 7: Since the authors use the Gabor filter instead of other common schemes to learn high-frequency signals, it would be good to have an ablation study on the Gabor filter part.
> >
> > Response: Thank you for your valuable suggestion. We agree that an ablation study on the Gabor filter part would provide further insights into its contribution to learning high-frequency signals. In response to your feedback, we will conduct additional experiments to assess the impact of the Gabor filter. The comprehensive results are now detailed in the revised paper, specifically in Section 4. High-level summaries of these findings are presented here (The evaluation metric used for performance measurement is Mean Squared Error): (1) The elimination of filters leads to a significant drop in model performance, highlighting the indispensability of filter designs; (2) Substituting the Gabor filter with a Fourier filter not only diminishes accuracy ($9.0\%$'s drop in simulation-based data and $18.1\%$'s drop in satellite-based data) but also increases the model size. This shows the adaptability of Gabor filters to different bandwidths for capturing patterns at various scales.
> >
> >
> > | Designs | $\#$ Param | Simulation |  Satellite |
> > | ----------- | ----------- | ----------- |  ----------- |
> > |  None | $577 K$ | $1.758e^{-2}$ |$6.883e^{-3}$ |
> > |  Fourier | $601 K$ | $4.439e^{-3}$ |$7.422e^{-4}$ |
> > |  Gabor | $581 K$ | $4.073e^{-3}$ |$6.290e^{-4}$ |

---

### Official Review · Reviewer_8E6g · 2023-10-26

**Soundness:** 3 good
**Presentation:** 3 good
**Contribution:** 3 good
**Rating:** 6
**Confidence:** 3

**Summary:**

Authors consider reconstruction of continuous fields from sparse data, which has important applications in many real world problem (as authors states). They propose a new approach based on implicit neural presentations (INRs) and apply it to two climate model related datasets.

**Strengths:**

Numerical results looks hood. Their methods seems to provide slower reconstruction accuracy (measured in terms of MSE) thank other INR approaches.

**Weaknesses:**

Presentation is quite a week and I have to say that I cannot follow it and understand what they are doing. First, mathematical notation is ambiguous and does not seem to follow standard mathematical principles, which creates confusion (Examples are in Questions section). Furthermore, background and description seems to be more like a list of different things and without clear connection. To make presentation more clear, authors could perhaps try to express their method with less mathematically rigours notations, but explaining their method with words (e.g. by providing example)

As mentioned above, numerical results looks good when compared to other INR methods. However, I would like to see also comparison to other non-INR based methods such as GP (if applicable) or methods that are currently used in meteorology and climate studies.

**Questions:**

Questions and comments:

- Problem setup: They assume that the system is governed by a PDE (1). I don't see any reason why this assumption is made, or is that assumption used somewhere?

- Section 2.1, "Objective" (and through the paper): They use notation $\mathcal U_k=(u(t_k,x)|\mathcal X_k)_{x\in X_k}$ . What this "$|\mathcal X_k$" stands in the set? $|$ usually is used for conditioning but I cannot see such a thing here. I would rather remove this extra part here to avoid confusion (I use '(' and ')' instead of curly brackets as this form does not seem to render those (and actually, strictly speaking, '(' and ')' would be preferred if order of measurements have meaning $\mathcal U_k$))

- The section 2.1 does not talk about noise or randomness and formulations in this section would indicate that $u(t,x)$ is deterministic. However, later sections seems to talk about noise and assume that $u$ is includes noise or is random (e.g. Eq (5) which does not have meaning if $u$ is deterministic). This is contradicting. I would suggest authors that authors would use separate notations for a noiseless quantity $u$ and the measurements, e.g., say something like $y_k$ is a vector of measurements or $u(t_k,x)$ plus noise.

- Section 2.3, "Neural pinpointing". They mention that learning-based methods can be computationally demanding. But there are two sides in this: train time computationally complexity and run time/inference computational complexity. GANs have high training complexity but inference may not be soo much, and perhaps even less as the proposed INR depending on the architectures.

- Section 2.4, "$\alpha=g(x)\circ \eta(t)$. The standard meaning for $\circ$ is composition of functions, i.e. $(f\circ g)(x)=f(g(x))$. Here this probable is not the case as I cannot see how you would compose $g$ and $\eta$ in this case. Of course, authors can define their own notations, but then they should say that how they use the notation. If $\circ$ is meant to be the multiplication, I would just remove $\circ$.

- Section 2.4, $\eta(t_k):\mathcal U_k=(u(t_k,x)|\mathcal X_k)_{x\in \mathcal X_k}\mapsto z_k$. Not sure what this is meant to mean. $\eta$ is a function which maps $t$ to a function which is a mapping between the values of $u$ to $z_k$?

- Section 3 seems to be a list of different things and it not clear to me that how the things are connected. For example, how you use auto-decoder? You have (auto-)decoder as encoder? How Gabor filters are used? Are these somehow connected to $g$ and $\eta$ earlier? Multiplicative filter network is decoder?

---

> ### Author Response · Authors · 2023-11-17
> **Response to Reviewer 8E6g**
>
> Dear reviewer, we appreciate your comments and acknowledge your concerns regarding our manuscript. We have carefully considered your feedback and made substantial revisions to enhance the clarity and coherence of our work. We have simplified the mathematical notations, providing more intuitive explanations and examples to facilitate better understanding. A simple walkthrough is provided below.
>
> **Problem** Our objective is to accurately reconstruct a spatiotemporal continuous physical field, denoted as $\boldsymbol{u}$, representing quantities such as temperature, velocity, or displacement. This field, $\boldsymbol{u}$, is inherently a function of both spatial coordinates ($\boldsymbol{x}$) and time ($t$).
>
> **Prior approaches**  Traditional implicit neural networks (INRs) incorporate the time index ($t$) along with spatial coordinates. For instance, in 2D data, the network features three input nodes for the $x$-coordinate, $y$-coordinate, and time $t$, where the time index ($t$) is employed to direct the model to a particular time instance.
>
> **Our approach** Motivated by the desire for a more context-aware indexing mechanism, we posed the question: Can the indexing process be improved? We propose utilizing measurements observed $u_t^{(1)}, u_t^{(2)}, \dots$ at time t for implicit model guidance, rather than relying on $t$ for explicit pointing. More precisely, our approach involves constructing an encoder $E_\varphi(\cdot)$ to transform the observed measurements $u_t^{(1)}, u_t^{(2)}, \dots$ into a latent code ($\boldsymbol{z}_t$). Subsequently, we employ a decoder $D_\phi(\cdot)$ that utilizes both the encoded information ($\boldsymbol{z}_t$) and spatial coordinates ($\boldsymbol{x}$) to reconstruct the underlying physical field. Below are detailed explanations of the model components we designed and the rationale behind each choice.
>
> - **Encoder**: The *auto-decoder* is chosen as the backbone for the encoder due to its ability to accommodate free-formed observations, considering that the number and positions of available observations vary over time.
> - **Gabor filter**: INRs often encounter challenges in learning high-frequency functions, a phenomenon known as "spectral bias." To mitigate this issue, we employ the Gabor filter to transform spatial coordinates ($\boldsymbol{x}$) before the decoding process.
> - **Decoder**: The decoder incorporates a multiplicative filter network (MFN) as its backbone network. This choice is motivated by the recursive mechanisms embedded in MFN, enabling improved fusion of spatial coordinates ($\boldsymbol{x}$) and latent codes ($\boldsymbol{z}$).
>
> **Results** We conducted comprehensive experiments to demonstrate the superior performance of the proposed model.
>
> In response to your feedback, we have extensively revised our paper, ensuring that the updated manuscript is more accessible and comprehensible. In the meantime, we provide point-to-point response to your comments below:
>
> ---
>
> Comment $\#$ 1: Problem setup: They assume that the system is governed by a PDE (1). I don't see any reason why this assumption is made, or is that assumption used somewhere?
>
> Response: We used PDE to facilitate the transition to functional variable separation and subsequently to our proposed context-aware indexing mechanism. However, we acknowledge the potential confusion it may have caused, and as a result, we have removed this equation from the paper.
>
> ---
>
> Comment $\#$ 2: Section 2.1, "Objective" (and through the paper): They use notation . What this " " stands in the set? usually is used for conditioning but I cannot see such a thing here. I would rather remove this extra part here to avoid confusion (I use '(' and ')' instead of curly brackets as this form does not seem to render those (and actually, strictly speaking, '(' and ')' would be preferred if order of measurements have meaning ))
>
> Response: Conditioning is employed in our notation because the latent codes are derived from all available position-dependent measurements at time $t$.
>
> ---
>
> Comment $\#$ 3: The section 2.1 does not talk about noise or randomness and formulations in this section would indicate that is deterministic. However, later sections seems to talk about noise and assume that is includes noise or is random (e.g. Eq (5) which does not have meaning if is deterministic). This is contradicting. I would suggest authors that authors would use separate notations for a noiseless quantity and the measurements, e.g., say something like is a vector of measurements or plus noise.
>
> Response: Thank you for your feedback. We have incorporated the suggested changes into the revised paper, and additional details regarding the model's performance under various noise levels have been included.

---

> > ### Author Response · Authors · 2023-11-17
> > **Response to Reviewer 8E6g**
> >
> > Comment $\#$ 4: Section 2.3, "Neural pinpointing". They mention that learning-based methods can be computationally demanding. But there are two sides in this: train time computationally complexity and run time/inference computational complexity. GANs have high training complexity but inference may not be soo much, and perhaps even less as the proposed INR depending on the architectures.
> >
> > Response: Yes, we agree with the reviewer. What we meant by "computationally demanding" was the challenge of adapting Generative Adversarial Networks (GANs) to irregular mesh and arbitrary-scale super-resolution settings.
> >
> > ---
> >
> > Comment $\#$ 5: Section 2.4, " . The standard meaning for is composition of functions, i.e. . Here this probable is not the case as I cannot see how you would compose and in this case. Of course, authors can define their own notations, but then they should say that how they use the notation. If is meant to be the multiplication, I would just remove.
> >
> > Response: Thank you for the comment. The introduction of the method has been rewritten for clarity, and a new figure has been added to facilitate a comparison between the proposed approach and current INRs.
> >
> > ---
> >
> > Comment $\#$ 6: Section 3 seems to be a list of different things and it not clear to me that how the things are connected. For example, how you use auto-decoder? You have (auto-)decoder as encoder? How Gabor filters are used? Are these somehow connected to and earlier? Multiplicative filter network is decoder?
> >
> > Response: Please refer to the previously provided walkthrough for detailed information.
> >
> > ---
> >
> > Comment $\#$ 7: As mentioned above, numerical results looks good when compared to other INR methods. However, I would like to see also comparison to other non-INR based methods such as GP (if applicable) or methods that are currently used in meteorology and climate studies.
> >
> > Response: We are actively working on implementing and comparing the suggested GP with MMGN. Once the experiments are concluded, we will share the results here.

---

> ### Comment · Reviewer_8E6g · 2023-11-21
> **Goor improvements, but not completely sure if enough**
>
> I read the revised manuscript. I admit that it fixed and clarified quite many parts of the original manuscript. But there are still some parts to improve and I am not confident that if the quality is enough for publication in ICLR.
>
> At least there points could be clarified and/or improved:
> - The key in their approach is the latent model $E$ which maps measurements from time $t$ to the latent variable, which is then fed to $D$ to get prediction of $u$ for the selected point $x$. This makes me to wonder that are measurements required for all time instants for which you want to infer $u$, meaning that the approach cannot be used to interpolate or extrapolate $u$ with respect to time? I also do not see mechanism how model could adapt to temporal behaviour of the physical system. If my interpretation is true, then the method acts more like a spatial interpolator instead of predicting a space-time system. This means that the model cannot be used, for example, for weather or climate prediction. I think this is a quite big limitation and should be addressed.  If this is not the case, I would wish to see an explanation that how the method is applied when $\mathcal U_t$ is empty.
>
> - There are some missing details such as $\mathcal T$ and the loss function are not specified. Appendix seems to include training details, but this section is not referenced in the main part of paper. Appendix also has some broken references (e.g below Eq (6)). These are not big things, but improves readability and I would not like to see such in published papers.
>
> - Numerical results look good, but baselines are mostly other INR methods. But I would as pointed out in the introduction, there are several other non-INR methods. I would wish to see those to be compared . Or if such comparison is done in another study, this could be referred.
>
> I increased my rating as this revision was a clear improvement, but still not sure if the manuscript is ready for publication.

---

> > ### Author Response · Authors · 2023-11-22
> > **Response to Reviewer 8E6g**
> >
> > Dear reviewer, thank you for your addtional comments and feedback. We have carefully considered them, and we are committed to addressing the your concerns to enhance the clarity and completeness of the paper.
> >
> > **Temporal Adaptability**
> > Temporal interpolation and extrapolation are closely connected to the core objective of our study, which revolves around field reconstruction rather than forecasting. While our focus remains on the reconstruction task, we have incorporated additional discussion on potential extensions to address forecasting scenarios based on the reviewer's comments:
> >
> > - **Reconstruction**: To investigate the reconstruction performance of the proposed model for time instances where $\mathcal{U}_t$​ is not available, we conducted additional experiments. During training, we masked certain time instances, for instance $[\mathcal{U}_1, \mathcal{U}_2, n/a, n/a,  $ $\mathcal{U}_5, \mathcal{U}_6, n/a,  $ $\mathcal{U}_8, n/a, \mathcal{U}_10]$. After training, we employed interpolation using latent codes, testing various interpolation methods. The following presents a comparison of different interpolation techniques.
> >
> >     | Method | Linear | Cubic | Spline |
> >     | ----------- | ----------- | ----------- | ----------- |
> >     |   Performance (MSE) | $4.159e^{-3}$ | $4.476e^{-3}$ | $4.082e^{-3}$ |
> >
> > - **Nowcasting**: Nowcasting typically involves predicting current or very near-future weather conditions within the next few hours, utilizing real-time observational data. In this context, we assume access to observations for the nowcasting model. During the inference stage, the model generates a new latent code based on these observations, while keeping all decoder parameters fixed.
> >
> >     | Task | Performance (MSE) |
> >     | ----------- | ----------- |
> >     |   Reconstruction | $3.997e^{-3}$ |
> >     |   Nowcasting | $4.012e^{-3}$ |
> >
> > - **Forecasting**: As the decoder remains fixed after training, the extrapolation for forecasting tasks involves predicting the latent code. This can be achieved through two types of methods: autoregressive methods and neural operator methods. Autoregressive methods iteratively update the latent code $\boldsymbol{z}(t+\Delta t)=\mathcal{M}(\Delta t, \boldsymbol{z}(t))$ where $\mathcal{M}: \mathbb{R} \times \mathbb{R}^{d_{\boldsymbol{z}}} \rightarrow \mathbb{R}^{d_{\boldsymbol{z}}}$ is the temporal update, while neural operator methods map a stack of latent codes to a future state/states: $\boldsymbol{z}_{t} = \mathcal{M}(t, \boldsymbol{z}_1, \boldsymbol{z}_2, \dots, \boldsymbol{z} (t-1))$. We are sorry that we have not completed the implementation within the available revision timeframe, but we will include this discussion in the revised paper for the benefit of researchers interested in these model extensions.
> >
> > **Broken References**:
> > We have rectified the broken references in the appendix and added definitions of T and loss function to enhance the overall readability of the paper. Specifically, observations are collected at $N_t$ consecutive time instances $\mathcal{T} $ and loss function is defined as the mean squared error (squared L2 norm) between the prediction $\hat{u}$ and target $u$.
> >
> >
> > **Comparison with Non-INR Methods**:
> > We conducted a comparison of our method with Gaussian processes. Specifically, we employed Gaussian process regression using scikit-learn, testing various kernels such as RBF, DotProduct, WhiteKernel, and their combinations. These experiments were conducted on simulation-based data, with 5% of the data points utilized for training. The Mean Squared Error (MSE) for our MMGN is $4.244e^{-3}$, and the performances of the GPs are provided below:
> >
> >
> > | Data size | 10 Instances |  20 Instances |  30 Instances |  40 Instances |  50 Instances |
> > | ----------- | ----------- | ----------- | ----------- | ----------- | ----------- |
> > |  GPs Performance | $7.76032e^{-2}$ | $6.86319e^{-2}$ | $7.26413e^{-2}$ | $6.26571e^{-2}$ | $5.31689e^{-2}$ |
> >
> > Overall, it's important to note that GPs come with a cubic cost in terms of dataset size. We conducted our experiments on an A100 GPU with 40 GB of memory, limiting the maximum number of time instances to less than 50, even with a low sampling rate. Additionally, we noticed that GPs are known to be sensitive to conditioning. Given the time constraints between the reviewer's comment update and the discussion period closing, we were only able to achieve the presented performance. However, we acknowledge that further improvements could be achieved with a more efficient implementation and hyperparameter tuning, although the computational cost remains high.

---

> > > ### Comment · Reviewer_8E6g · 2023-11-22
> > >
> > > Sounds good to me. I raised my score, assuming that these suggested revisions will be delivered to final version.
> > >
> > > Of course, not all aspects of temporal adaptivity needs to be considered in this paper. But it is important that mention possible meaninful limitations and perhaps discuss about possible extension/ideas of the work for future. Btw I am wondering if ML approaches such as RNNs could be applied to to predict $z$ between or after measuremens?

---

> > > > ### Author Response · Authors · 2023-11-22
> > > > **Response to Reviewer 8E6g**
> > > >
> > > > Dear reviewer, thank you for considering the revisions to your evaluation. We are actively working on incorporating the feedback from the second round of comments and questions into the final version of the paper. Concerning the application of ML approaches such as RNNs for predicting between or after measurements, it is an interesting point; it belongs to the autoregressive methods group we mentioned in the forecast setting.

---

### Official Review · Reviewer_A6YW · 2023-11-02

**Soundness:** 3 good
**Presentation:** 3 good
**Contribution:** 3 good
**Rating:** 6
**Confidence:** 5

**Summary:**

This paper proposes an INR–based method to reconstruct continuous fields from sparse sensor data for the application of climate modeling. The proposed method mainly introduces a latent variable to capture temporal information. The experiments are conducted on both simulated and real data.

**Strengths:**

- The paper is well written and organized, which is easy to follow.
- The motivation and target application of the proposed method is clear and convincing.
- The proposed method introduces a latent vector z to incorporate with INR decoder, which sounds interesting since it seems to make the INR model can be generalized across different images/samples by changing or optimizing latent vectors.
- The experiments are conducted on both simulated data and satellite-based data with a clear comparison with baseline methods.

**Weaknesses:**

- About latent code z. The paper claims to use the trainable latent code to represent temporal information instead of using time index. What is the motivation for this? Because the recorded time index is not reliable in this application? Or the latent code can also represent other semantic information? Previous works have shown INR can represent spatial-temporal information in a good way by adding temporal index as input coordinates.

- As described in the paper, during training, both latent code and INR are trained. While during training, INR model parameters are fixed and only latent code is optimized. Since the given information is quite sparse during testing, how reliable it can be to optimize latent code? How robust this method can be when generalized across different objects? By only changing the latent code in the input of INR decoder and freezing all other model weights, can this change the decoded image in an efficient way?

- For the comparison with baseline methods as in Table 1, it seems all the baseline methods use a different encoding than the proposed model, which uses Gabor filters over Fourier bases to transform the coordinates as Fourier transforms emphasize a global frequency representation. Usually, the INR models are quite sensitive with different encoding methods. Can the paper also report the comparison with baseline methods using the same Gabor filters as encoding to disclose how impactful the encoding method is? Similar with the backbone model architecture, are all the comparisons based on the same backbone model? Only in this way, it may be fair to say how efficient the latent code can be in this case.

- The compared methods are all the object-specific or scene-specific models, while the proposed model seems to be a generalized model across multiple images or objects. The method may also need to be compared with: (1) other generalized models based on INR such as SRN [1], etc. (2) other spatial-temporal models using time index such as D-NeRF [2].

- The visualization of latent code is quite interesting but it is not clear what it means for the Figure 5. Can the author explain more about what information is learned or embedded by the latent code from these plots?

[1] Scene Representation Networks: Continuous 3D-Structure-Aware Neural Scene Representations, NeurIPS 2019.

[2] ​​D-NeRF: Neural Radiance Fields for Dynamic Scenes, CVPR 2021.

**Questions:**

Please see weakness for the details of questions.

---

> ### Author Response · Authors · 2023-11-17
> **Response to Reviewer A6YW**
>
> Dear reviewer, we are grateful for your comments and questions. We find them very helpful! Particularly those pertaining to our model designs and the proposed ablation study on the Gabor filter's effects. Overall, we have revised the methodology section to elucidate the distinctions between the current INR and our approach, providing explanations for our design choices. Please find our detailed responses below.
>
> ---
>
> Comment $\#$ 1: About latent code z. The paper claims to use the trainable latent code to represent temporal information instead of using time index. What is the motivation for this? Because the recorded time index is not reliable in this application? Or the latent code can also represent other semantic information? Previous works have shown INR can represent spatial-temporal information in a good way by adding temporal index as input coordinates.
>
> Response: Yes, the latent code has the advantage of capturing not only temporal information but also other semantic features of the underlying physical process. Previous studies have indeed demonstrated the capability of INR to effectively capture spatialtemporal information by incorporating a temporal index as additional input coordinates. But it should be noted that different from forecasting problems, the time index ($t$) is primarily used as a reference to indicate a specific time instance in the context of field reconstruction. Motivated by the desire for a more **context-aware indexing mechanism**, we introduce a new perspective to the modeling process by utilizing a trainable latent code. Since the latent code is learned using the available measurements at time $t$ via auto-decoding, it is a more context-aware representation, which improves the performance of the INR-based decoder. We have expanded the methodology section to elaborate on these points, providing a detailed discussion on the motivation behind our model design choices and a visual comparison between the traditional INR and our proposed approach. We sincerely invite the reviewer to check the updated paper and provide any further feedback or suggestions you may have.
>
> ---
>
> Comment $\#$ 2: As described in the paper, during training, both latent code and INR are trained. While during training, INR model parameters are fixed and only latent code is optimized. Since the given information is quite sparse during testing, how reliable it can be to optimize latent code? How robust this method can be when generalized across different objects? By only changing the latent code in the input of INR decoder and freezing all other model weights, can this change the decoded image in an efficient way?
>
> Response: Thank you for the insightful question regarding the generalizability of latent codes. To investigate this aspect, we conducted additional experiments by partitioning the simulation-based data set into two subsets, each comprising 512 instances. The temporal splitting was performed in an ascending order, with the initial 512 instances assigned to subset $\mathcal{D}_1$ and the subsequent 512 instances to subset $\mathcal{D}_2$. Consequently, due to extrapolation in the time direction, the distributions of $\mathcal{D}_1$ and $\mathcal{D}_2$ could vary significantly. We trained MMGN using $\mathcal{D}_1$. Then, keeping the decoder parameters fixed, we performed inference on $\mathcal{D}_2$, where the only trainable parameters were the newly initialized latent codes. The results indicate a marginal increase in error, demonstrating that the proposed MMGN avoids the catastrophic forgetting issue encountered by other time-index-based INR methods when re-trained using different data set.
>
> | Data set | MSE |
> | ----------- | ----------- |
> |   $\mathcal{D}_1$ | $3.997e^{-3}$ |
> |   $\mathcal{D}_2$ | $4.012e^{-3}$ |
>
> Admittedly, one limitation of our current work is its focus on reconstructing a selected trajectory. In the future, we plan to investigate MMGN's potential for generalization to multiple trajectories or even across various climate properties, with the goal of recovering arbitrary underlying flow maps. We have incorporated this discussion into the conclusion section of the revised paper.

---

> > ### Author Response · Authors · 2023-11-17
> > **Response to Reviewer A6YW**
> >
> > Comment $\#$ 3: For the comparison with baseline methods as in Table 1, it seems all the baseline methods use a different encoding than the proposed model, which uses Gabor filters over Fourier bases to transform the coordinates as Fourier transforms emphasize a global frequency representation. Usually, the INR models are quite sensitive with different encoding methods. Can the paper also report the comparison with baseline methods using the same Gabor filters as encoding to disclose how impactful the encoding method is? Similar with the backbone model architecture, are all the comparisons based on the same backbone model? Only in this way, it may be fair to say how efficient the latent code can be in this case.
> >
> > Response: We appreciate the reviewer's suggestion regarding the encoding method and backbone model architecture in our comparison with baseline methods. To further study the impact of the Gabor filter, we conducted additional ablation experiments, encompassing the removal of filter designs and their replacement with other types. The comprehensive results are now detailed in the revised paper, specifically in Section 4. High-level summaries of these findings are presented here (The evaluation metric used for performance measurement is Mean Squared Error): (1) The elimination of filters leads to a significant drop in model performance, highlighting the indispensability of filter designs; (2) Substituting the Gabor filter with a Fourier filter not only diminishes accuracy ($9.0\%$'s drop in simulation-based data and $18.1\%$'s drop in satellite-based data) but also increases the model size. This shows the adaptability of Gabor filters to different bandwidths for capturing patterns at various scales.
> >
> > | Designs | $\#$ Param | Simulation |  Satellite |
> > | ----------- | ----------- | ----------- |  ----------- |
> > |  None | $577 K$ | $1.758e^{-2}$ |$6.883e^{-3}$ |
> > |  Fourier | $601 K$ | $4.439e^{-3}$ |$7.422e^{-4}$ |
> > |  Gabor | $581 K$ | $4.073e^{-3}$ |$6.290e^{-4}$ |
> >
> > ---
> >
> > Comment $\#$ 4: The compared methods are all the object-specific or scene-specific models, while the proposed model seems to be a generalized model across multiple images or objects. The method may also need to be compared with: (1) other generalized models based on INR such as SRN [1], etc. (2) other spatial-temporal models using time index such as D-NeRF [2].
> >
> > Response: Thank you for your valuable suggestion. We appreciate your insightful comments and tried our best to implement the suggested comparisons. It is still a work in progress. In the current revision, we have included some discussion in the related work section, acknowledging the importance of these comparisons.
> >
> > ---
> >
> > Comment $\#$ 5: The visualization of latent code is quite interesting but it is not clear what it means for the Figure 5. Can the author explain more about what information is learned or embedded by the latent code from these plots?
> >
> > Response: Thank you for the question. We acknowledge the need for improved clarity in our presentation, and we have refined the descriptions in the revised paper accordingly.

---

> > > ### Comment · Reviewer_A6YW · 2023-11-23
> > >
> > > Thank you so much for the author to answering my questions, especially with new experiments comparison and ablation study are conducted. Most of my concerns have been addressed.
> > >
> > > For #3: How to understand this table when compared with the main Table 1? E.g. when other baselines are using the same encoding as the proposed MMGN, how to compare with the results in a more fair way?
> > >
> > > For #4: Thanks for the author' efforts. I understand it may be quite challenging to finish all these experiments in the time limited rebuttal period. But I think these methods should be more significant and relevant baselines to the setting in this paper with latent variable and dynamic scenes.
> > >
> > > I have also read the other reviewer's comments. Overall, I will keep my original rating.

---

### Official Review · Reviewer_MqB4 · 2023-11-05

**Soundness:** 3 good
**Presentation:** 2 fair
**Contribution:** 2 fair
**Rating:** 6
**Confidence:** 3

**Summary:**

The authors propose a new implicit representation for spatio--temporal fields. The representation is based on separately processing temporal information via an encoder--decoder architecture (the decoder is an auto-decoder). This strategy, combined with other ingredients such as Fourier (here Gabor) features and multiplicative filter networks yields strong empirical performance on a synthetic climate dataset.

**Strengths:**

- the authors skillfully combine several existing techniques into a new architecture

- compared with the baselines presented in the paper, the proposed method achieves remarkably strong performance

- the idea to use an auto-decoder is creative and nice

- the paper is well illustrated (for example, Figure 2 is very helpful in understanding the architecture)

**Weaknesses:**

- I find the paper very hard to read. Notation and expressions in Section 3 and the last paragraph of Section 2 are confusing and it takes a long time to understand what is going on (Figure 2 helps here but the math could be clearer). Another reason is very florid prose which I'd hesitate to use in a machine learning paper. Examples: "The endeavor to tackle the conundrum of global field reconstruction from sparse observations is currently shaping two distinct paradigms of thought", "raging ocean waves", "intricate physical fields" (what are they?), "we present an _innovative_ neural network approach", "... a singular viable option", ...

- While the narrative insists on strong novelty, all ingredients (and various combination thereof) are well known. The separation of variable idea is based on Dona et al 21, multiplicative filter networks are known, local activations (such as Gabor or wavelet) have been used in many papers, e.g. https://openaccess.thecvf.com/content/CVPR2023/html/Saragadam_WIRE_Wavelet_Implicit_Neural_Representations_CVPR_2023_paper.html. In some sense the present paper is a multiplicative filter network with a separate treatment of time through an encoder--decoder. This is perfectly fine but in my view it limits the novelty, and there is no shortage of all sorts of implicit networks.

- In real fields space and time _are_ intertwined so one needs to go beyond the rank-one decomposition f(x) g(t); Dona et al mentions a low rank version \sum_k f_k(x) g_k(t). It is not clear to me from the current description how this required expresiveness is achieved. It would be nice to explain that more clearly.

- Generally, there is a lack of insight on why this strategy is supposed to perform well and why other approaches are supposed to perform poorly. This is a more general problem: there are now hundreds of implicit networks with different hyperparameters, training strategies, etc, and very little theoretical or other understanding of their mechanics.
What if you compare your approach with an optimally tuned adaptive kernel estimator? How about spline interpolation? How about a simple low pass filter? It will surely not mess up as badly as ResMLP in Figure 3.

- Along the same lines, in datasets description you mention that wavelets were used for initial interpolation of satellite-based data on a grid; why not compare the described method with a sparse wavelet interpolation?

**Questions:**

- in experiments where you recover fields on the surface of a sphere, are you using Cartesian or spherical polar coordinates?

- why not include (different levels of) random noise in observations? this is an important test of robustness

- I am not sure that (Izacard et al., 2019) is the right reference for sparse seismic networks and small earthquakes (or even for sparse spatial coverage in scientific data; it addresses a super-resolution problem)

---

> ### Author Response · Authors · 2023-11-17
> **Response to Reviewer MqB4**
>
> We appreciate the valuable insights offered by the reviewer, particularly the recommendation to conduct additional experiments involving varying levels of noise to enhance the demonstration of the model's robustness. Additionally, we are thankful for the thoughtful and practical suggestions for revising the presentation of the paper. Subsequently, we present our responses to each comment in a point-by-point manner.
>
> ---
>
> Comment $\#$ 1: in experiments where you recover fields on the surface of a sphere, are you using Cartesian or spherical polar coordinates?
>
> Response: In experiments involving the recovery of fields on the surface of a sphere, we utilize spherical polar coordinates for modeling.
>
> ---
>
> Comment $\#$ 2: why not include (different levels of) random noise in observations? this is an important test of robustness.
>
> Response: Thank you for the suggestion. We have conducted additional experiments considering the introduction of noise to the input data. The noise level in the dataset is quantified by the channelwise standard deviation specific to that dataset. Furthermore, we introduced customized noise ratios, including scenarios with noise levels set at 1\%, 5\%, and 10\%. The quantitative results are presented below. Additionally, we made a new figure to enhance the visualization of performance distinctions among various models in the presence of noise. The detailed discussion and presentation of these findings have been incorporated into the revised paper.
>
> | Model | Noise free | $1\%$ | $5\%$ | $10\%$ |
> | ----------- | ----------- | ----------- | ----------- | ----------- |
> |  ResMLP | $0.01852$ | $0.01895$ | $0.01969$ | $0.02214$ |
> |  FFN+P  | $0.02357$ | $0.02463$ | $0.03020$ | $0.04324$ |
> |  FFN+G  | $0.02982$ | $0.03239$ | $0.03774$ | $0.05335$ |
> |  MMGN   | $0.00448$ | $0.00452$ | $0.00629$ | $0.01387$ |
>
> ---
>
> Comment $\#$ 3: I am not sure that (Izacard et al., 2019) is the right reference for sparse seismic networks and small earthquakes (or even for sparse spatial coverage in scientific data; it addresses a super-resolution problem). I find the paper very hard to read. Notation and expressions in Section 3 and the last paragraph of Section 2 are confusing and it takes a long time to understand what is going on (Figure 2 helps here but the math could be clearer). Another reason is very florid prose which I'd hesitate to use in a machine learning paper. Examples: "The endeavor to tackle the conundrum of global field reconstruction from sparse observations is currently shaping two distinct paradigms of thought", "raging ocean waves", "intricate physical fields" (what are they?), "we present an innovative neural network approach", "... a singular viable option", ...
>
> Response: We thank for the valuable feedback. We have made several improvements to the paper. Specifically, we corrected the reference in the introduction and provided a clearer explanation of the contributions. Moreover, we enhanced the methodology section by introducing the model more explicitly and illustrating the design process as well as adding a new figure.

---

> > ### Author Response · Authors · 2023-11-17
> > **Response to Reviewer MqB4**
> >
> > Comment $\#$ 4: While the narrative insists on strong novelty, all ingredients (and various combination thereof) are well known. The separation of variable idea is based on Dona et al 21, multiplicative filter networks are known, local activations (such as Gabor or wavelet) have been used in many papers, e.g. Wavelet Implicit Neural Representations. In some sense the present paper is a multiplicative filter network with a separate treatment of time through an encoder--decoder. This is perfectly fine but in my view it limits the novelty, and there is no shortage of all sorts of implicit networks.
> >
> > Response: We truly appreciate this comment, recognizing that our initial presentation lacked clarity regarding the proposed model. We acknowledge the multifaceted nature of the model, as highlighted by the reviewer. In response, our revised paper offers a more explicit discussion, elucidating the rationale behind the selection of each component. Notably, we emphasize the pivotal innovation introduced in our work: the **context-aware indexing mechanism**.
> >
> > Previous Implicit Neural Representation (INR) models incorporated a temporal index ($t$) as additional input coordinates. Different from scenarios involving forecasting, where the time index plays a more active role, in the context of field reconstruction, we argue it primarily serves as a reference for indicating a specific time instance. Consequently, our aim is to devise an improved pointing method. The latent codes, derived from available context information through the utilization of measurements from the underlying physical process at time t, encapsulate richer semantic information. This is exemplified in extreme cases, such as fixing the latent size to 1, where our proposed MMGN consistently outperforms other INR baselines, as demonstrated in the following experimental results.
> >
> > | Model | MSE (Simulation-based data) |  MSE (Satellite-based data) |
> > | ----------- | ----------- | ----------- |
> > |  ResMLP | $0.01950$ | $0.00154$ |
> > |  FFN+P |  $0.02516$ | $0.00303$ |
> > |  FFN+G |  $0.03239$ | $0.00510$ |
> > |  SIREN |  $0.27304$ | $0.31291$ |
> > |  MMGN (latent size = 1) |   $\boldsymbol{0.01898}$ | $\boldsymbol{0.00152}$ |
> >
> > ---
> >
> > Comment $\#$ 5: In real fields space and time are intertwined so one needs to go beyond the rank-one decomposition f(x) g(t); Dona et al mentions a low rank version sum k f-k(x) g-k(t). It is not clear to me from the current description how this required expresiveness is achieved. It would be nice to explain that more clearly.
> >
> > Response: Yes, achieving a clear separation of spatial function $f(x)$ and temporal function $f(x)$ in modeling poses significant challenges. In our approach, we reconsider the roles of spatial coordinates $x$ and temporal index $t$. Broadly, the time index $t$ serves as a means to guide the model to a specific time instance. In field reconstruction, we can (1) leverage $t$ for explicit pointing or (2) use measurements observed at time $t$ for implicit model guidance. We adopted the second approch. Consequently, there is no explicit need for variable separation.

---

> > > ### Author Response · Authors · 2023-11-17
> > > **Response to Reviewer MqB4**
> > >
> > > Comment $\#$ 6: Generally, there is a lack of insight on why this strategy is supposed to perform well and why other approaches are supposed to perform poorly. This is a more general problem: there are now hundreds of implicit networks with different hyperparameters, training strategies, etc, and very little theoretical or other understanding of their mechanics. What if you compare your approach with an optimally tuned adaptive kernel estimator? How about spline interpolation? How about a simple low pass filter? It will surely not mess up as badly as ResMLP in Figure 3.
> > >
> > > Response: We conducted supplementary experiments to examine the enhancements facilitated by the context-aware indexing mechanism, as demonstrated in the extreme case results presented earlier. Additionally, we explored the effects of the Gabor filter through additional experiments, the outcomes of which are detailed below. Our observation of MMGN's superior performance is attributed to the synergistic impact of the context-aware indexing mechanism and the Gabor filter-based decoder utilized in our model.
> > >
> > > | Designs | $\#$ Param | Simulation |  Satellite |
> > > | ----------- | ----------- | ----------- |  ----------- |
> > > |  None | $577 K$ | $1.758e^{-2}$ |$6.883e^{-3}$ |
> > > |  Fourier | $601 K$ | $4.439e^{-3}$ |$7.422e^{-4}$ |
> > > |  Gabor | $581 K$ | $4.073e^{-3}$ |$6.290e^{-4}$ |
> > >
> > > ---
> > >
> > > Comment $\#$ 7: Along the same lines, in datasets description you mention that wavelets were used for initial interpolation of satellite-based data on a grid; why not compare the described method with a sparse wavelet interpolation?
> > >
> > > Response: The second dataset utilized in our study is sourced from GHRSST (Group for High-Resolution Sea Surface Temperature). Various data levels are available, with higher levels signifying increased spatial coverage. In the initial stages of our project, we aimed for greater data control to assess different INR models. Consequently, we opted for level 4, wherein the input data has already undergone preprocessing via sparse wavelet interpolation.

---

> > > > ### Comment · Reviewer_MqB4 · 2023-11-21
> > > >
> > > > I thank the authors for earnestly engaging with my comments. I would increase my score to a 6 since the revision is indeed an improvement over the original submission. However, Reviewer 8E6g raised an important concern about the ability of the proposed model to learn / interpolate temporal dynamics which I would first like to see addressed.

---

> > > > > ### Author Response · Authors · 2023-11-22
> > > > > **Response to Reviewer MqB4**
> > > > >
> > > > > Dear Reviewer, thank you for acknowledging the improvements in our revised submission. We have addressed the second round of comments and questions raised by Reviewer 8E6g. We encourage you to review these updates, and we are open to any further questions or suggestions you may have.

---

> > > > > > ### Comment · Reviewer_MqB4 · 2023-11-22
> > > > > >
> > > > > > Thank you. I updated the score.

---

### Author Response · Authors · 2023-11-17
**Overall Response**

First of all, we appreciate all reviewers for their valuable insights. It is encouraging to note the consensus among reviewers on the significance of the field reconstruction problem in various scientific applications and the superiority of our proposed model over current INR baselines. In response to the constructive comments, we have carefully revised the paper. In this summary, we offer an overview of the major changes and address common concerns. Subsequently, we will provide individual responses to each reviewer, offering further details on the revisions.

- **Additional experiments**: In response to the reviewer's comments, we conducted thorough additional experiments, enhancing the paper from the following five key aspects:
    - **Convergence study** Additional experiments were conducted to assess the convergence at extremely low sampling rates. The analysis was further connected with proper orthogonal decomposition to investigate and validate the sampling limit.
    - **Robustness to noise** Additional experiments considering the introduction of noise to the input data have been conducted. Results have been incorporated into the revised paper's results section.
    - **Ablations of Gabor filter** Two types of experiments were conducted: First, we substitute the proposed Gabor filter with alternative designs. Secondly, we remove filters altogether. The computed performance metrics illustrate the contribution made by the Gabor filter.
    - **Ablations of context-aware indexing mechanism** We reduced the latent size to 1 and conducted experiments to showcase the efficacy of the proposed context-aware indexing mechanism ($z$) in comparison to the conventional time indexing ($t$).
    - **Generalizability of latent codes** We conducted supplementary experiments to illustrate that the proposed MMGN does not face the challenges of catastrophic forgetting commonly observed in existing INR models.

- **Paper presentation**: We have reworked different sections of the paper.
    - **Introduction** We have revised the introduction and emphasized the machine learning problem of interest and highlighted our introduction of the **context-aware indexing mechanism** to INR modeling.
    - **Methodology** We have updated the methodology section to highlight the novel modeling perspective we introduced compared to existing INR models. Additionally, we have provided a thorough explanation and motivation for our approach, supported by a new figure for clarity.
    - **Results** We have incorporated the suggested experiments discussed above into the revised version of the paper: three new paragraphs, two new figures, and one new table. Additionally, we have redesigned some figures to enhance the presentation of the XAI analysis results, eliminating potential confusion.
    - **Conclusion** We have added limitations and future work to the conclusion section.

---

> ### Author Response · Authors · 2023-11-17
> **Overall Response**
>
> Dear reviewers, we've incorporated feedback into our revised manuscript. Your thoughtful input has been very helpful. We kindly invite you to reassess the paper, updating your comments and scores if possible. Thank you for your time and effort.

---

> > ### Author Response · Authors · 2023-11-21
> > **Overall Response**
> >
> > Dear reviewer, we would like to bring to your attention that it has been four days since we submitted our revised manuscript, and with only one day remaining in the reviewer-author discussion period, we are eager to hear your feedback. If you could reevaluate our revised paper to assess if our response has adequately addressed your concerns, or if you have any further questions, please let us know. Your valuable input is highly appreciated, and we look forward to your response.

---

### Meta-Review · Area_Chair_KTkf · 2023-12-08

**Metareview:**

This submission presents an implicit representation (INR) for modelling physics data, where information on a continous field is reconstructed from sparse recontructions. The 4 expert reviewers generally appreciated

- a clear motivation,
- the idea of an additional latent code for the temporal information as input to the INR,
- the idea of Gabor filtering for learning high-frequency signals,
- numerical performance,
- visual presentation.

The reviews also brought up that the paper is in essence a combination of existing techniques, the somewhat limited complexity of the target datasets and lacking ablations.
The authors' answers could clear up some of the concerns, in particular by introducing additional ablations, for instance on the positional encoddings, and scores were raised.

The AC judges that in spite of the weaknesses the paper is of value to the field and recommends acceptance.

P.S. The AC asks the authors to respect paper margins and in particular to fit Table 1 into the required space.

**Justification For Why Not Higher Score:**

Given the weaknesses, there is no reason to accept the paper as a spotlight.

**Justification For Why Not Lower Score:**

There is no clear reason to reject this paper unless wants to more insist on the limited scope of the datasets, but which are standard in the community.

---

### Decision · Program_Chairs · 2024-01-16

Accept (poster)